# Lipschitz-Guided Design of Interpolation Schedules in Generative Models

## Abstract

We study the design of interpolation schedules in flow and diffusion-based generative models from both statistical and numerical perspectives. Within the stochastic interpolants framework, we first show that scalar interpolation schedules are statistically equivalent under the Kullback–Leibler divergence in path space, after optimal a posteriori tuning of the diffusion coefficient. This equivalence motivates focusing on numerical properties of the drift field rather than purely statistical criteria. We propose minimizing the averaged squared Lipschitzness of the drift as a principled criterion for schedule design, in contrast with kinetic-energy minimization in optimal transport. A simple transfer formula expresses the drift of one schedule in terms of the drift of another, allowing the designed schedule to be used at inference time with a model trained under a different (e.g., linear) schedule, without retraining. We work out the optimal schedules analytically for Gaussian and Gaussian-mixture targets: for Gaussians, we obtain exponential improvements in the Lipschitz constant over linear schedules; for Gaussian mixtures, we obtain schedules that mitigate mode collapse in few-step sampling. We then validate the approach on high-dimensional invariant measures of stochastic Allen–Cahn and Navier–Stokes equations, where the designed schedule yields markedly more accurate fine-scale statistics at fixed integrator budget.

## Contents

# 1 Introduction

## 1.1 Context

Dynamics between probability measures, particularly flows and diffusion processes described by ordinary and stochastic differential equations (ODEs and SDEs), underpin many modern generative modeling techniques Sohl-Dickstein et al. (2015); Ho et al. (2020); Song et al. (2020). These models generate samples by progressively transforming simple noise into structured data through a sequence of intermediate distributions, implemented as an iterative denoising or refinement process Song & Ermon (2019); Karras et al. (2022). The manner in which these transformations are traversed in time—encoded by interpolation or noise schedules—plays a crucial role in shaping the resulting generative dynamics and their performance.

In this paper, we study the mathematical design of interpolation schedules for flow and diffusion-based generative models within the stochastic interpolants framework Albergo et al. (2023); Albergo & Vanden-Eijnden (2022). This framework provides a unified formulation for noising and denoising processes based on sample interpolation, and enables principled construction of the associated generative dynamics. It is closely related to concurrent developments such as flow matching Lipman et al. (2022) and rectified flows Liu et al. (2022), and encompasses diffusion and score-based generative models Sohl-Dickstein et al. (2015); Song et al. (2020); Ho et al. (2020); Song & Ermon (2020) as special cases.

## 1.2 Basics of stochastic interpolants

Let $x_1 \sim \mu^*$, where $\mu^*$ is a target probability supported on $\mathbb{R}^d$ satisfying $\mathbb{E}[\|x_1\|_2^2] = \int_{\mathbb{R}^d} \|x\|_2^2 \mu^*(\mathrm{d}x) < \infty$. The linear stochastic interpolant with scalar schedule is the stochastic process $I_t = \alpha_t z + \beta_t x_1$, where $z \sim \mathsf{N}(0, \mathrm{I})$ is multivariate normal distributed with $z \perp x_1$. Here $\alpha_t, \beta_t \in C^1([0, 1])$ are scalar functions of $t$ satisfying the boundary conditions $\alpha_0 = \beta_1 = 1$ and $\alpha_1 = \beta_0 = 0$, so that $I_0 = z$ and $I_1 = x_1$.

For different values of $t$, the interpolant $I_t$ can be seen as modeling a corruption of the target at a specific scale. The theory of stochastic interpolants Albergo & Vanden-Eijnden (2022); Albergo et al. (2023) shows that one can generate samples from $\mu^*$ by solving the following ODE:

$$\mathrm{d}X_t = b_t(X_t)\mathrm{d}t, \quad X_0 \sim \mathsf{N}(0, \mathrm{I}), \tag{1.1}$$

where $b_t(x) = \mathbb{E}[\dot{I}_t | I_t = x]$ and $\dot{I}_t$ denotes the time derivative of $I_t$. The solution satisfies $\mathrm{Law}(X_t) = \mathrm{Law}(I_t)$, and in particular, $X_1 \sim \mu^*$. This can also be seen as a consequence of the mimicking theorem Gyöngy (1986), also referred to as Markovian projection.

Because the drift $b_t$ is a conditional expectation, we can define it as the minimizer of the square loss function

$$L(\hat{b}) = \int_0^1 \mathbb{E}[\|\hat{b}_t(I_t) - \dot{I}_t\|_2^2] \, \mathrm{d}t \,.$$

By parametrizing $\hat{b}$ in an expressive class, using e.g. a deep neural network, and optimizing the loss function (with expectation over empirical samples), we obtain an approximation $\hat{b} \approx b$. This allows us to solve (1.1) with $\hat{b}_t$ to generate samples. More technical details and variants for SDEs and the *a posteriori* tuning of diffusion coefficients are presented in Section 2.1.

## 1.3 This work

Since $\hat{b}_t$ is learned from samples and generation requires numerical integration of a differential equation, a natural question arises: does there exist a particular choice of $\alpha_t, \beta_t$ that can enhance both statistical and numerical efficiency? This paper establishes design principles for addressing this question. Specifically, our contributions are as follows:

- In Section 2, we prove that under the Kullback-Leibler divergence criterion in path space, different choices of scalar schedules are *statistically equivalent* when diffusion coefficients are optimized a posteriori. This equivalence fundamentally renders statistical considerations insufficient for schedule selection.

- In Sections 3.1 and 3.2, we propose minimizing the averaged squared Lipschitzness of the drift $b_t$ as a principled criterion for schedule design. A *transfer formula* (Proposition 3.1) expresses the drift of one scalar schedule in terms of the drift of another, so the designed schedule can be used at inference time with a model trained under a different schedule, without retraining.

- In Sections 3.3–3.5, we work out the optimal schedules analytically for Gaussian and Gaussian-mixture targets. For Gaussians, the designed schedule yields *exponential* improvements in the Lipschitz constant of the drift; for Gaussian mixtures, it mitigates mode collapse under few-step sampling. We also extend the construction to log-concave and general distributions.

- In Section 4, we validate the design on high-dimensional Gaussian fields and mixtures, and on invariant measures of stochastic Allen–Cahn and Navier–Stokes equations. The designed schedule, applied at inference time via the transfer formula, produces more accurate enstrophy spectra at fixed integrator budget.

## 1.4 Related work

Since the introduction of flow and diffusion-based generative models, a large body of work has explored their design principles and parameter choices; see the survey Yang et al. (2023). These studies address, among other aspects, the choice of noise distributions, noising and denoising processes, time-reversal dynamics, training objectives, and diffusion coefficients. The present work focuses on the design of interpolation schedules within the unit-time stochastic interpolants framework Albergo & Vanden-Eijnden (2022); Albergo et al. (2023), which relates to concurrent developments in flow matching Lipman et al. (2022) and rectified flows Liu et al. (2022), and encompasses diffusion and score-based generative models Sohl-Dickstein et al. (2015); Song et al. (2020); Ho et al. (2020); Song & Ermon (2020) as special cases. Within this framework, interpolation schedules play a role analogous to noise schedules in diffusion models.

Schedule design has been studied from both statistical and numerical perspectives. From a statistical standpoint, Kingma et al. (2021) showed that different noise schedules in diffusion models yield the same variational lower bound. Our results indicate that this notion of *statistical equivalence* extends to a broader setting within the unit-time stochastic interpolants framework when the Kullback–Leibler divergence in path space is used as the estimation criterion; see Remark 2.6. This observation suggests that statistical considerations alone are insufficient to distinguish among scalar interpolation schedules.

From a numerical perspective, most existing work has relied on empirical studies—primarily on machine learning benchmarks—to tune noise or time schedules for improved sampling efficiency San-Roman et al. (2021); Jolicoeur-Martineau et al. (2021); Nichol & Dhariwal (2021); Song & Ermon (2020); Karras et al. (2022). Related approaches learn improved schedules or time parametrizations using additional training Shaul et al. (2024); Xue et al. (2024); Sabour et al. (2024); Chen et al. (2024a), while Wang et al. (2024) analyzes the sensitivity of schedule choice to score estimation errors. Here we propose a principled approach to numerical schedule design based on optimizing the Lipschitz regularity of the drift field at inference time, without requiring retraining.

Related mathematical work has investigated Lipschitz regularity, contractivity, and stability properties of flows and flow maps Daniels (2025); Tsimpos et al. (2025). In particular, Tsimpos et al. (2025) considers a variational problem for minimizing the maximum Lipschitz constant under reparametrization of a fixed flow, a setting which differs from ours. See also Aranguri et al. (2025) for a mathematical study of how schedule design affects mode identification in high-dimensional distributions.

Another closely related line of work advocates learning optimal transport paths Liu et al. (2022); Pooladian et al. (2023), which are straight and therefore appealing from a numerical standpoint; related models based on entropy-regularized optimal transport include Schrödinger bridges De Bortoli et al. (2021); Shi et al. (2023); Pooladian & Niles-Weed (2025). However, optimal transport paths may give rise to highly irregular drift fields Tsimpos et al. (2025), which are not well suited for numerical integration (see also Remark 3.4). Moreover, since the estimation of the initial drift $b_0$ inevitably contains errors, one-step or few-step generation strategies that rely heavily on straightness may struggle to accurately reproduce fine-scale features. Our

proposed Lipschitz-based criterion is instead designed to directly target numerical integration efficiency and robustness to discretization.

Finally, numerical efficiency can also be improved through advances orthogonal to schedule design, including higher-order, exponential, and parallel integrators Dockhorn et al. (2022); Lu et al. (2022); Zhang & Chen (2022); Li et al. (2024); Chen et al. (2024b); Wu et al. (2024); De Bortoli et al. (2025); Tan et al. (2025), as well as multiscale and cascading approaches Yu et al. (2020); Dhariwal & Nichol (2021); Jing et al. (2022); Saharia et al. (2022); Ho et al. (2022); Guth et al. (2022); Phung et al. (2023). In addition, consistency models and approaches based on learning flow maps Song et al. (2023); Kim et al. (2024); Salimans & Ho (2022); Frans et al. (2024); Boffi et al. (2025) aim to reduce the number of sampling steps altogether. These approaches are complementary to schedule design and can be combined with the methods studied in this work.

We note that interpolation schedules can also substantially influence training stability and efficiency. The present paper focuses primarily on the statistical and numerical efficiency of schedule design.

## 2 Statistical Equivalence under Kullback-Leibler in Path Space

In this section, we discuss the statistical properties of different interpolation schedules, using the Kullback-Leibler (KL) divergence in path space as the criterion. The focus is on formal derivations and calculations, and the goal is to reveal the underlying structures rather than provide a fully rigorous treatment, which would require delicate discussions on the regularity of the SDEs.

### 2.1 Stochastic interpolants

Here we briefly recall the main results of the stochastic interpolant framework Albergo & Vanden-Eijnden (2022); Albergo et al. (2023). For completeness, we also include a simple sketch of derivations in Appendix A.

As in Section 1.2, we denote the target distribution by $\mu^*$, and assume that it is supported on $\mathbb{R}^d$ and satisfies $\mathbb{E}[\|x_1\|_2^2] < \infty$. For simplicity we also assume that $\mu^*$ is absolutely continuous with respect to the Lebesgue measure and has a smooth density.

**Definition 2.1.** *The linear stochastic interpolant between $x_1 \sim \mu^*$ and the Gaussian noise $z \sim \mathsf{N}(0, \mathrm{I})$ with $z \perp x_1$ is the process*

$$I_t = \alpha_t z + \beta_t x_1, \quad 0 \le t \le 1. \tag{2.1}$$

*where $\alpha_t, \beta_t \in C^1([0,1])$ are scalar interpolation schedules satisfying the boundary conditions $\alpha_0 = \beta_1 = 1$ and $\alpha_1 = \beta_0 = 0$ as well as $\dot{\beta}_t > 0, \dot{\alpha}_t < 0$ for $t \in (0,1)$.*

The law of the stochastic interpolant coincide with the law of the solution of an ODE with a drift given by a conditional expectation:

**Proposition 2.2.** *Let $b_t(x) = \mathbb{E}[\dot{I}_t | I_t = x]$. Then the solutions to the ODE*

$$\mathrm{d}X_t = b_t(X_t)\mathrm{d}t, \quad X_0 \sim \mathsf{N}(0, \mathrm{I}),$$

*satisfy $\mathrm{Law}(X_t) = \mathrm{Law}(I_t)$ for all $t \in [0,1]$, and in particular, $X_1 \sim \mu^*$.*

Using the Fokker-Planck equation and the fact that $\nabla \cdot (\rho \nabla \log \rho) = \Delta \rho$, we can also construct a family of SDEs that share the same law at each time as the interpolation process $I_t$:

**Proposition 2.3.** *Let $b_t(x) = \mathbb{E}[\dot{I}_t | I_t = x]$ and assume the density of $I_t$, denoted by $\rho_t$, exists and is $C^1$ in space. Then for any $\epsilon_t \ge 0$, the solutions to the SDE*

$$\mathrm{d}X_t = (b_t(X_t) + \epsilon_t \nabla \log \rho_t(X_t))\, \mathrm{d}t + \sqrt{2\epsilon_t}\mathrm{d}W_t, \quad X_0 \sim \mathsf{N}(0, \mathrm{I}).$$

*satisfy $\mathrm{Law}(X_t) = \mathrm{Law}(I_t)$ for all $t \in [0,1]$, and in particular, $X_1 \sim \mu^*$.*

By Stein's identity, the score $\nabla \log \rho_t(x)$ can be expressed as:

$$\nabla \log \rho_t(x) = -\frac{1}{\alpha_t}\mathbb{E}[z | I_t = x]. \tag{2.2}$$

By using

$$x = \mathbb{E}[I_t | I_t = x] = \alpha_t \mathbb{E}[x_0 | I_t = x] + \beta_t \mathbb{E}[x_1 | I_t = x]$$
$$b_t(x) = \mathbb{E}[\dot{I}_t | I_t = x] = \dot{\alpha}_t \mathbb{E}[x_0 | I_t = x] + \dot{\beta}_t \mathbb{E}[x_1 | I_t = x],$$

(2.3)

after some simple algebra we can relate $b_t(x)$ and $\nabla \log \rho_t(x)$ through an affine transformation

$$b_t(x) = \frac{\dot{\beta}_t}{\beta_t} x + \alpha_t^2 (\frac{\dot{\beta}_t}{\beta_t} - \frac{\dot{\alpha}_t}{\alpha_t}) \nabla \log \rho_t(x).$$

(2.4)

This means that, if we know $b_t$ or an approximation of it, we can use the above relation to obtain the score or an approximation of it directly.

## 2.2 Learning the drift from data

We can use empirical risk minimization to learn the conditional expectation $b$ through optimizing the square loss function

$$L(\hat{b}) = \int_0^1 \mathbb{E}[\|\hat{b}_t(I_t) - \dot{I}_t\|_2^2] \, dt.$$

In practice, the expectation is over empirical samples. Optimizing it leads to an estimate of $\hat{b}$.

It is also common to optimize for the denoiser $\mathbb{E}[x_1 | I_t = x]$, or the score $\nabla \log \rho_t(x) = -\mathbb{E}[\frac{z}{\alpha_t} | I_t = x]$ directly. The corresponding loss functions can be similarly constructed since these terms are all expressed as conditional expectations. We note that the three objects can be recovered from each other by affine transformations, using (2.3) and (2.4). Thus, without loss of generality and for a unified analysis, let us assume that at the end we have an estimator of the score in terms of $\hat{s}_t(x) \approx \nabla \log \rho_t(x)$. This means that the estimated SDE has the form

$$d\hat{X}_t = \left( \frac{\dot{\beta}_t}{\beta_t} \hat{X}_t + (\alpha_t^2 (\frac{\dot{\beta}_t}{\beta_t} - \frac{\dot{\alpha}_t}{\alpha_t}) + \epsilon_t) \hat{s}_t(x) \right) dt + \sqrt{2\epsilon_t} dW_t, \quad \hat{X}_0 \sim \mathsf{N}(0, \mathrm{I}).$$

## 2.3 Optimizing the KL in path space

Given the flexibility of choosing $\epsilon_t$, it is natural to ask which $\epsilon_t$ is optimal. Let us consider the criterion of the KL divergence between path measures $\mathbb{P}_X$ and $\mathbb{P}_{\hat{X}}$ of $X = (X_t)_{0 \le t \le 1}$ and $\hat{X} = (\hat{X}_t)_{0 \le t \le 1}$, respectively. According to Girsanov's theorem, this KL divergence has the form

$$\mathrm{KL}[\mathbb{P}_X \| \mathbb{P}_{\hat{X}}] = \frac{1}{2\epsilon_t} \int_{\mathbb{R}^d} \int_0^1 \left( \alpha_t^2 (\frac{\dot{\beta}_t}{\beta_t} - \frac{\dot{\alpha}_t}{\alpha_t}) + \epsilon_t \right)^2 \|\nabla \log \rho_t(x) - \hat{s}_t(x)\|_2^2 \rho_t(x) dt dx.$$

(2.5)

Now, recall the fact that, for any $a$, the minimizer of $\frac{(\epsilon + a)^2}{2\epsilon} = \frac{\epsilon}{2} + a + \frac{a^2}{2\epsilon}$ is $\epsilon = |a|$, and the minimum is $\max\{0, 2a\}$. Thus, the KL achieves minimum when $\epsilon_t = \alpha_t^2 (\frac{\dot{\beta}_t}{\beta_t} - \frac{\dot{\alpha}_t}{\alpha_t})$. Viewing this optimized KL as a function of the interpolation schedules $\alpha, \beta$ and denoting it as $\mathrm{KL}^\star(\alpha, \beta)$, it reads

$$\mathrm{KL}^\star(\alpha, \beta) = 2 \int_{\mathbb{R}^d} \int_0^1 \alpha_t^2 (\frac{\dot{\beta}_t}{\beta_t} - \frac{\dot{\alpha}_t}{\alpha_t}) \|\nabla \log \rho_t(x) - \hat{s}_t(x)\|_2^2 \rho_t(x) dt dx.$$

(2.6)

*Remark* 2.4. For certain choices of $\alpha_t, \beta_t$, the resulting $\epsilon_t$ may blow up. However, the SDE is still well defined; see examples in Appendix B. ◇

## 2.4 Equivalence over scalar schedules

Our next result shows that, remarkably, $\mathrm{KL}^\star(\alpha, \beta)$ remains constant regardless of the interpolation schedules $\alpha_t, \beta_t$ we choose.

**Proposition 2.5.** *Let $q_\eta(x)$ be the probability density function of $x_1 + \eta z$ with $\eta \geq 0$ and denote by $\hat{S}_\eta(x)$ an estimator of its score $\nabla \log q_\eta(x)$ derived from $\hat{s}_t(x)$. Then*

$$\mathrm{KL}^\star(\alpha, \beta) = 2 \int_0^\infty \eta \cdot \mathbb{E}[\|\nabla \log q_r(x_1 + \eta z) - \hat{S}_r(x_1 + \eta z)\|_2^2]\mathrm{d}\eta. \tag{2.7}$$

*Proof.* We know that $\rho_t(x)$ is the density of $\alpha_t z + \beta_t x_1 = \beta_t(x_1 + \frac{\alpha_t}{\beta_t}z)$. Thus $\nabla \log \rho_t(x) = \frac{1}{\beta_t}\nabla \log q_{\frac{\alpha_t}{\beta_t}}(\frac{x}{\beta_t})$, and $\hat{s}_t(x) = \frac{1}{\beta_t}\hat{S}_{\frac{\alpha_t}{\beta_t}}(\frac{x}{\beta_t})$. Using these relations, we have

$$\begin{aligned}
\mathrm{KL}^\star(\alpha, \beta) &= 2\int_{\mathbb{R}^d}\int_0^1 \frac{\alpha_t^2}{\beta_t^2}\left(\frac{\dot{\beta}_t}{\beta_t} - \frac{\dot{\alpha}_t}{\alpha_t}\right)\|\nabla \log q_{\frac{\alpha_t}{\beta_t}}(\frac{x}{\beta_t}) - \hat{S}_{\frac{\alpha_t}{\beta_t}}(\frac{x}{\beta_t})\|_2^2 \rho_t(x)\mathrm{d}t\mathrm{d}x \\
&= 2\int_0^1 \frac{\alpha_t^2}{\beta_t^2}\left(\frac{\dot{\beta}_t}{\beta_t} - \frac{\dot{\alpha}_t}{\alpha_t}\right)\mathbb{E}[\|\nabla \log q_{\frac{\alpha_t}{\beta_t}}(x_1 + \frac{\alpha_t}{\beta_t}z) - \hat{S}_{\frac{\alpha_t}{\beta_t}}(x_1 + \frac{\alpha_t}{\beta_t}z)\|_2^2]\mathrm{d}t.
\end{aligned} \tag{2.8}$$

Noting that $\frac{\alpha_t^2}{\beta_t^2}(\frac{\dot{\beta}_t}{\beta_t} - \frac{\dot{\alpha}_t}{\alpha_t}) = -\frac{\alpha_t}{\beta_t}\frac{\mathrm{d}}{\mathrm{d}t}(\frac{\alpha_t}{\beta_t})$ and using $\alpha_t/\beta_t$ instead of $t$ as integration variable, we arrive at (2.7). $\qquad\square$

*Remark* 2.6. In Kingma et al. (2021), it was pointed out that in diffusion models, different noise schedules lead to the same variational lower bound. In the continuous setting, this corresponds to the KL divergence in path space. Our results generalize their discussion to stochastic interpolants and incorporate the step of a posteriori tuning of diffusion coefficients. $\qquad\diamond$

Proposition 2.5 shows that the optimal KL accuracy in path space depends solely on the estimation of $\nabla \log q_r(x_1 + rz)$: that is, from the perspective of KL divergence in path space, all linear scalar interpolants with independently coupled endpoints and one endpoint being Gaussian are statistically indistinguishable. This indicates that other metrics need to be explored if we want to select models for improved statistical efficiency. On the other hand, using nonlinear or matrix-valued instead of scalar schedules may potentially lead to different statistical efficiency, a direction of interest in future work.

# 3 Numerical Design by Optimizing Averaged Squared Lipschitzness

The previous section established that all scalar interpolation schedules are statistically equivalent under KL in path space. What this equivalence does *not* touch is numerical efficiency: schedules that are statistically interchangeable can produce ODEs whose drift fields differ dramatically in regularity, and hence in how easily they can be integrated to high accuracy with few time steps. In this section we propose a numerical criterion – the averaged squared Lipschitzness of the drift – and study its minimization over scalar schedules. We focus on ODEs rather than SDEs for simplicity, noting that ODEs typically achieve better empirical performance due to their greater ease of integration Karras et al. (2022); Dockhorn et al. (2022).

## 3.1 From one schedule to another: a transfer formula

We first observe that the drift of any scalar schedule can be expressed in closed form in terms of the drift of any other scalar schedule. This *transfer formula* is what allows the schedule designs we develop later in this section to be used at inference time with a model trained under a different schedule, without retraining. Without loss of generality, we take as reference the linear schedule $\alpha_t^\dagger = 1 - t$, $\beta_t^\dagger = t$.

**Proposition 3.1** (Transfer formula)**.** *Consider the two stochastic interpolants $I_t^\dagger = \alpha_t^\dagger z + \beta_t^\dagger x_1$ and $I_t = \alpha_t z + \beta_t x_1$ and their associated drifts $b_t^\dagger(x) = \mathbb{E}[\dot{I}_t^\dagger|I_t = x]$ and $b_t(x) = \mathbb{E}[\dot{I}_t|I_t = x]$. Then with $t^\dagger = 1/(1 + \alpha_t/\beta_t)$, it holds that*

$$b_t(x) = \frac{\dot{\alpha}_t}{\alpha_t}x + \left(\dot{\beta}_t - \frac{\dot{\alpha}_t\beta_t}{\alpha_t}\right)\left((1 - t^\dagger)b_{t^\dagger}^\dagger\left(\frac{t^\dagger}{\beta_t}x\right) + \frac{t^\dagger}{\beta_t}x\right). \tag{3.1}$$

*Proof.* By direct algebraic calculations, we get

$$
\begin{aligned}
b_t^\dagger(x) &= \mathbb{E}[x_1 - z | I_t = x] = \mathbb{E}[x_1 - \frac{I_t - t x_1}{1 - t} | I_t = x] \\
&= -\frac{x}{1-t} + \frac{1}{1-t}\mathbb{E}[x_1 | x_1 + \frac{1-t}{t} z = \frac{x}{t}],
\end{aligned}
\tag{3.2}
$$

and similarly

$$
\begin{aligned}
b_t(x) &= \mathbb{E}[\dot\alpha_t z + \dot\beta_t x_1 | I_t = x] = \mathbb{E}[\dot\alpha_t \frac{I_t - \beta_t x_1}{\alpha_t} + \dot\beta_t x_1 | I_t = x] \\
&= \frac{\dot\alpha_t}{\alpha_t} x + (\dot\beta_t - \frac{\dot\alpha_t \beta_t}{\alpha_t})\mathbb{E}[x_1 | x_1 + \frac{\alpha_t}{\beta_t} z = \frac{x}{\beta_t}].
\end{aligned}
\tag{3.3}
$$

Let $t^\dagger$ satisfy $\alpha_t/\beta_t = (1 - t^\dagger)/t^\dagger$. This means that $t^\dagger = 1/(1 + \alpha_t/\beta_t)$. Therefore, combining (3.2) and (3.3), we arrive at (3.1). $\qquad\square$

The proposition implies that we can change the interpolation schedule from one to another whenever we know the drift function for any reference schedule. The same identity applies to learned estimators of the drift, so the schedule can be tuned *at inference time* rather than during training. Related identities have appeared in the literature Kingma et al. (2021); Karras et al. (2022). We use the transfer formula systematically in the experiments of Section 4: every model is trained with the linear schedule, and the designed schedule is applied at inference by composing the trained drift through Proposition 3.1. The remaining question is how to choose the new schedule, which we address next.

## 3.2 Optimizing averaged squared Lipschitzness

As natural and principled approach to choose the schedule, we propose to minimize the following averaged squared Lipschitzness criterion.

**Definition 3.2.** *The averaged squared Lipschitzness (avg-Lip$^2$) is defined as*

$$
A_2 = \int_0^1 \mathbb{E}[\|\nabla b_t(I_t)\|_2^2]\, \mathrm{d}t,
\tag{3.4}
$$

*where $\|\cdot\|_2$ is the 2-norm.*

In general, we could optimize $A_2$ over all possible nonlinear interpolants $I_t$. Here, for simplicity, we restrict ourselves to linear interpolants with scalar schedules $\alpha, \beta$[1]. We provide several examples in the next two sections and show the significance of this criterion in numerical performance and compares it with optimal transport.

## 3.3 1D example: Gaussian

We begin with analytic studies on 1D Gaussians.

**Example 3.3** (1D Gaussian). *Consider $I_t = \alpha_t z + \beta_t x_1$ with $x_1 \sim \mathsf{N}(0, M) \perp z \sim \mathsf{N}(0, 1)$. Here $M > 0$ is a positive scalar. Then*

$$
b_t(x) = \mathbb{E}[\dot I_t | I_t = x] = \mathrm{Cov}(\dot I_t, I_t)\mathrm{Cov}(I_t)^{-1}x = (\alpha_t\dot\alpha_t + \beta_t\dot\beta_t M)(\alpha_t^2 + \beta_t^2 M)^{-1}x.
$$

*If we take $\alpha_t = 1 - t, \beta_t = t$, we get*

$$
b_t(x) = \frac{t - 1 + tM}{(1-t)^2 + t^2 M}x.
$$

*Suppose $M$ is a large number[2]. We have*

$$
A_2 = \int_0^1 \frac{(t - 1 + tM)^2}{((1-t)^2 + t^2 M)^2}\mathrm{d}t \geq \int_{\frac{1}{M^{1/3}}}^{\frac{1}{M^{1/2}}} \frac{(t - 1 + tM)^2}{((1-t)^2 + t^2 M)^2}\mathrm{d}t \geq \Omega(\sqrt{M}).
$$

---

[1]See discussions on matrix-valued schedules in Remark 3.10.

[2]Although we can always use variance preserving design to fix this setting, it may still occur for a particular Fourier frequency component in high high-dimensional setting. Similar discussions apply when $M$ is a small number.

*Moreover, the Lipschitzness $\|\nabla b_t(1/M)\|_2 \geq \Omega(M)$ which grows linearly with $M$.*

*However, we can optimize*

$$A_2 = \int_0^1 \mathbb{E}[\|\nabla b_t(I_t)\|_2^2]\,\mathrm{d}t = \int_0^1 \mathbb{E}[\|\mathrm{Cov}(\dot{I}_t, I_t)\mathrm{Cov}(I_t)^{-1}\|_2^2]\,\mathrm{d}t$$
$$= \frac{1}{4}\int_0^1 \left\|\frac{\mathrm{d}}{\mathrm{d}t}\log\mathrm{Cov}(I_t)\right\|_2^2 \mathrm{d}t\,. \tag{3.5}$$

*By Cauchy–Schwarz inequality, the minimizer satisfies $\frac{\mathrm{d}}{\mathrm{d}t}\log\mathrm{Cov}(I_t) = \mathrm{const}$. To achieve the minimum, we get $\log\mathrm{Cov}(I_t) = (1-t)\log\mathrm{Cov}(I_0) + t\log\mathrm{Cov}(I_1)$. Solving this equation yields $\alpha_t^2 + \beta_t^2 M = M^t$. Taking the choice $\alpha_t^2 = 1 - \beta_t^2$, we obtain the interpolation schedule*

$$\alpha_t = \sqrt{\frac{M - M^t}{M - 1}}, \beta_t = \sqrt{\frac{M^t - 1}{M - 1}}\,. \tag{3.6}$$

*For such choice, $b_t(x) = \frac{1}{2}(\log M)x$. The corresponding $A_2 = O(\log^2 M)$ and $\|\nabla b_t(x)\|_2 \leq \frac{1}{2}|\log M|$ for all $t \in [0, 1], x \in \mathbb{R}$. This shows that there is an exponential improvement in the averaged squared Lipschitzness and the actual Lipschitz constant of the drift, compared to $\alpha_t = 1 - t, \beta_t = t$.*

*Remark 3.4. We compare the above to optimal transport, which minimizes the squared path length $P = \int_0^1 \mathbb{E}[\|b_t(I_t)\|_2^2]\,\mathrm{d}t$. Using the optimal transport theory[3], we get that*

$$b_t(x) = \frac{\sqrt{M} - 1}{1 - t + t\sqrt{M}}x\,.$$

*This can have a large Lipschitz constant near $t = 0$ when $M$ is large.* ◇

### 3.4 1D example: Gaussian mixture

We then move to Gaussian mixture.

**Example 3.5** (1D Gaussian mixtures)**.** *Consider the 1D bimodal Gaussian mixture*

$$\mu^*(x) = p\mathsf{N}(x; M, 1) + (1 - p)\mathsf{N}(x; -M, 1)\,.$$

*To enable an explicit analytic study[4], we take $\alpha_t = \sqrt{1 - \beta_t^2}$, which leads to*

$$b_t(x) = \dot{\beta}_t M\tanh(h + \beta_t Mx)\,, \tag{3.7}$$

*where $h$ satisfies $\frac{p}{1-p} = \exp(2h)$, or equivalently $p = \frac{\exp(h)}{\exp(h) + \exp(-h)}$.*

*Suppose $h > 0$. If $\beta_t = t$ and $M$ is large, we observe that at the initial time, $b_0(x) = M\tanh(h)$, which is large. In the one-dimensional case, this means all points move toward the right when using a forward Euler discretization with step size $O(1)$. Even for negative $x$, such a drift will likely push these points into positive territory. On the other hand, we know that for $x > 0$, we have $b_t(x) > 0$. This means that once a point reaches the positive side, it will remain positive. Therefore, such a discretization scheme will miss the mode on the left side. The above argument demonstrates that we must use an initial step size of $O(1/M)$ to ensure that the discretization does not miss modes.*

*Below, we study the optimization of avg-Lip$^2$, which leads to a schedule $\beta$ that grows slowly at initial time that does not suffer from the mode missing issue, namely, we can safely use a discretization scheme with uniform stepsize.*

---

[3]See details in Appendix C.1.
[4]See calculation details in Remark C.3 in Appendix C.

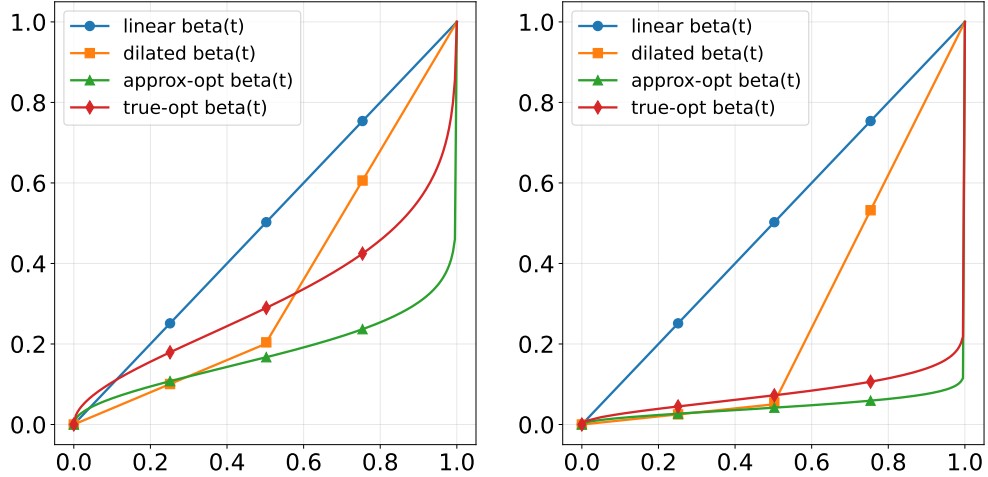

Figure 1: Comparison of different interpolation schedules $\beta_t$. Left: $M = 5$. Right: $M = 20$. We set $p = 0.3$. For the dilated schedule, we take $\kappa = 1$.

**Proposition 3.6** (Optimizing avg-Lip$^2$ for 1D Gaussian mixture)**.** *For the 1D bimodal Gaussian mixture example, if we optimize $A_2$ over all possible linear interpolants $I_t$ with scalar schedules satisfying $\alpha_t^2 + \beta_t^2 = 1$, then the optimal $\beta_t$ ($0 \le t \le 1$) satisfies*

$$t = \frac{\int_0^{\beta_t} u(G(u))^{1/2}\mathrm{d}u}{\int_0^1 u(G(u))^{1/2}\mathrm{d}u}\,, \tag{3.8}$$

*where $G(u) = \mathbb{E}[\mathrm{sech}^4(h + uM(\sqrt{1-u^2}z + ux_1))]$. Equivalently, we have the following Euler-Lagrange equation for the optimal $\beta_t$:*

$$-\dot{\beta}_t^2\beta_t - \ddot{\beta}_t\beta_t^2 + 2\dot{\beta}_t^2\beta_t^3 M^2(1 + \frac{3}{4}\,\mathrm{Corr}(I_t\tanh(h + \beta_t MI_t), \mathrm{sech}^4(h + \beta_t MI_t))) = 0\,,$$

*where $I_t = \sqrt{1-\beta_t^2}z + \beta_t x_1$. If we omit the Corr term, we get $\dot{\beta}_t^2\beta_t - \ddot{\beta}_t\beta_t^2 + 2\dot{\beta}_t^2\beta_t^2 M^2 = 0$ which has the solution*

$$\beta_t = \frac{1}{M}\sqrt{-\log(1 + (e^{-M^2} - 1)t)}\,. \tag{3.9}$$

*The proof of this proposition is in Appendix C.3.*

*Remark* 3.7. The time-dilated schedule studied in Aranguri et al. (2025) also resolve the mode missing issue:

$$\beta_t = \begin{cases} \dfrac{2\kappa t}{M}, & t \in \left[0, \frac{1}{2}\right], \\ \dfrac{\kappa}{M} + \left(1 - \dfrac{\kappa}{M}\right)(2t - 1), & t \in \left[\frac{1}{2}, 1\right]. \end{cases} \tag{3.10}$$

where $\kappa$ is a constant. $\diamondsuit$

We plot different schedules in Figure 1 and we solve for the true solutions numerically using (3.8). The dilated (3.10), optimal min-avg-Lip$^2$ (3.8), and approximate min-avg-Lip$^2$ solution (3.9) all exhibit slower growth near $t = 0$ compared to the standard linear schedule. Their key difference lies in their behavior near $t = 1$. The optimal and approximate min-avg-Lip$^2$ solutions exhibit more rapid growth near $t = 1$, which may cause numerical issues. However, their initial slowness allows the method to sample both modes without using a small stepsize, as we demonstrate in Section 4.2.

*Remark* 3.8. One may optimize instead $\int_0^1 \mathbb{E}[\|\nabla b_t(I_t)\|_2^{2k}]\,\mathrm{d}t$, then the optimal $\beta_t$ will satisfy

$$t = \frac{\int_0^{\beta_t} u(G(u))^{1/2k}\mathrm{d}u}{\int_0^1 u(G(u))^{1/2k}\mathrm{d}u}\,,$$

where now $G(u) = \mathbb{E}[\text{sech}^{4k}(h + uM(\sqrt{1 - u^2}z + ux_1))]$ and a similar ODE for $\beta_t$ holds. For details, see Appendix C.3. Detailed investigation of choice of $k$ is out of the scope of this paper, which may improve the behavior near $t = 1$. $\diamond$

### 3.5 High dimensional examples

We then move beyond 1D examples.

**Proposition 3.9** (Optimizing avg-Lip$^2$ for high dimensional Gaussians). *Consider $x_1 \sim \mathsf{N}(0, M) \perp z \sim \mathsf{N}(0, \mathrm{I})$ in $d$ dimensions with $M$ now a positive-definite symmetric matrix. Denote the eigendecomposition $M = U\Lambda U^T$ where $U$ is an orthogonal matrix and $\Lambda = \text{diag}(\lambda^{(1)}, ..., \lambda^{(d)})$ with $1 \geq \lambda^{(1)} \geq \lambda^{(2)} \geq ... \geq \lambda^{(d)} > 0$.*

*If we optimize $A_2$ over all possible linear interpolants $I_t$ with scalar schedules, then, the optimal solution is $I_t = \alpha_t z + \beta_t x_1$ with*

$$\alpha_t = \sqrt{\frac{\lambda^\star - (\lambda^\star)^t}{\lambda^\star - 1}}, \beta_t = \sqrt{\frac{(\lambda^\star)^t - 1}{\lambda^\star - 1}}. \tag{3.11}$$

*where $\lambda^\star = \lambda^{(d)}$. For the optimal solution, the corresponding 2-norm $\|\nabla b_t(x)\|_2 = \frac{1}{2}|\log \lambda^\star|$.*

*Proof of Proposition 3.9.* First, because the interpolant is linear and $z, x_1$ are jointly Gaussian, we have that $I_t, \dot{I}_t$ are jointly Gaussian. Thus,

$$b_t(x) = \mathbb{E}[\dot{I}_t | I_t = x] = \text{Cov}(\dot{I}_t, I_t)\text{Cov}(I_t)^{-1}x = (\alpha_t\dot{\alpha}_t + \beta_t\dot{\beta}_t M)(\alpha_t^2 + \beta_t^2 M)^{-1}x.$$

We can calculate the 2-norm using the eigenvalues:

$$\|\nabla b_t(x)\|_2 = \max_{1 \leq j \leq d}\left|\frac{\alpha_t\dot{\alpha}_t + \beta_t\dot{\beta}_t\lambda^{(j)}}{\alpha_t^2 + \beta_t^2\lambda^{(j)}}\right| = \max\left\{\left|\frac{\alpha_t\dot{\alpha}_t + \beta_t\dot{\beta}_t\lambda^{(1)}}{\alpha_t^2 + \beta_t^2\lambda^{(1)}}\right|, \left|\frac{\alpha_t\dot{\alpha}_t + \beta_t\dot{\beta}_t\lambda^{(d)}}{\alpha_t^2 + \beta_t^2\lambda^{(d)}}\right|\right\},$$

where, in the last equality, we used the fact that the function $\lambda \to \frac{\alpha_t\dot{\alpha}_t + \beta_t\dot{\beta}_t\lambda}{\alpha_t^2 + \beta_t^2\lambda}$ is non-decreasing. This implies that for $\lambda = \lambda^{(1)}$ or $\lambda^{(d)}$,

$$A_2 = \int_0^1 \mathbb{E}[\|\nabla b_t(I_t)\|_2^2]\mathrm{d}t \geq \int_0^1 \left|\frac{\alpha_t\dot{\alpha}_t + \beta_t\dot{\beta}_t\lambda}{\alpha_t^2 + \beta_t^2\lambda}\right|^2 \mathrm{d}t = \frac{1}{4}\int_0^1\left|\frac{\mathrm{d}}{\mathrm{d}t}\log(\alpha_t^2 + \beta_t^2\lambda)\right|^2\mathrm{d}t.$$

By Cauchy–Schwarz inequality, $A_2 \geq \frac{1}{4}\log^2\lambda$ for $\lambda = \lambda^{(1)}$ or $\lambda^{(d)}$. Using the assumption and definition $\lambda^\star$, we have $A_2 \geq \frac{1}{4}\log^2\lambda^\star$. Similar to the discussion in Section 3.3, the minimum can be achieved by taking $\frac{\mathrm{d}}{\mathrm{d}t}\log(\alpha_t^2 + \beta_t^2\lambda^\star) = \log\lambda^\star$; the assumption $1 \geq \lambda^{(1)}$ is used to verify the minimum. Taking $\alpha_t = \sqrt{1 - \beta_t^2}$ then leads to the solution in (3.11). $\square$

Proposition 3.9 shows that by adapting the interpolation schedules, the Lipschitz constant of the drift field depends on the magnitude of eigenvalues logarithmically, compared to algebraically when using the simple schedule $\alpha_t = 1 - t, \beta_t = t$. This is similar to the discussion for the 1D case in Section 3.3. For non-Gaussian targets where the eigenvalues of $M$ are not directly available, the parameter $\lambda^\star$ can be set from a Gaussian reference: in our experiments on stochastic PDE invariant measures (Section 4.4), we use the ratio between the data and noise spectra at the finest resolved frequency as a data-driven proxy for $\lambda^\star$.

*Remark* 3.10 (Discussions on matrix-valued schedules). If we allow matrix-valued schedules, it is possible to further improve numerical efficiency by adapting the schedule to each eigenvalue individually. In detail, consider the following choice:

$$\alpha_t = U\text{diag}(\alpha_t^{(1)}, \ldots, \alpha_t^{(d)})U^T, \quad \beta_t = U\text{diag}(\beta_t^{(1)}, \ldots, \beta_t^{(d)})U^T,$$

where

$$\alpha_t^{(k)} = \sqrt{\frac{\lambda^{(k)} - (\lambda^{(k)})^t}{\lambda^{(k)} - 1}}, \quad \beta_t^{(k)} = \sqrt{\frac{(\lambda^{(k)})^t - 1}{\lambda^{(k)} - 1}}.$$

When $\lambda^{(k)} = 1$, we interpret this formula through the limit $\lambda^{(k)} \to 1$. Direct calculation using this formula yields

$$
\begin{aligned}
b_t(x) = \mathrm{Cov}(\dot{I}_t, I_t)\mathrm{Cov}(I_t)^{-1}x &= (\dot{\alpha}_t\alpha_t^T + \dot{\beta}_t M\beta_t^T)(\alpha_t\alpha_t^T + \beta_t M\beta_t^T)^{-1}x \\
&= \frac{1}{2}U\mathrm{diag}(\log\lambda^{(1)}, \ldots, \log\lambda^{(d)})U^T x\,.
\end{aligned}
$$

Here, each eigenvector direction corresponds to its individual Lipschitz constant $|\log\lambda^{(i)}|$ for $1 \leq i \leq d$, and not all scales suffer from the largest $|\log\lambda^\star|$. We leave the investigation of matrix-valued schedules for future study. $\diamond$

**Example 3.11** (Extension to log-concave distributions)**.** *We can generalize the discussion of high-dimensional Gaussians to log-concave distributions. Let $\mu^* \propto \exp(-V)$ with $V \in C^2(\mathbb{R}^d)$ and $\lambda_m I \preceq \nabla^2 V \preceq \lambda_M I$ where we assume $\lambda_m \geq 1$. Consider $x_1 \sim \mu^*$ independent of $z \sim \mathsf{N}(0, \mathrm{I})$. Then for the linear interpolant with scalar schedule $I_t = \alpha_t z + \beta_t x_1$, we have*

$$
\frac{\alpha_t\dot{\alpha}_t + \beta_t\dot{\beta}_t\lambda_M^{-1}}{\alpha_t^2 + \beta_t^2\lambda_M^{-1}} \preceq \nabla b_t(x) \preceq \frac{\alpha_t\dot{\alpha}_t + \beta_t\dot{\beta}_t\lambda_m^{-1}}{\alpha_t^2 + \beta_t^2\lambda_m^{-1}}\,.
$$

*This can be proved using the Cramér–Rao and Brascamp–Lieb inequalities; see Gao et al. (2024). Therefore, similar to the Gaussian case, we can choose $\lambda^\star = \lambda_M^{-1}$. Then, with the schedule*

$$
\alpha_t = \sqrt{\frac{\lambda^\star - (\lambda^\star)^t}{\lambda^\star - 1}}, \quad \beta_t = \sqrt{\frac{(\lambda^\star)^t - 1}{\lambda^\star - 1}}\,, \tag{3.12}
$$

*we have $\|\nabla b_t(x)\|_2 \leq \frac{1}{2}|\log\lambda^\star|$. In general, we do not know an explicit solution for optimizing $A_2$ for log-concave distributions. However, the above schedule serves as a good choice, and the bound is tight and yields the optimal $A_2$ when the log-concave distribution is Gaussian.*

**Example 3.12** (A particular example on high dimensional Gaussian mixtures)**.** *Consider the bimodal Gaussian mixture in $d$ dimensions*

$$
\mu^*(x) = p\mathsf{N}(x; r, \mathrm{I}) + (1 - p)\mathsf{N}(x; -r, \mathrm{I})\,, \tag{3.13}
$$

*where $x \in \mathbb{R}^d$, and $r \in \mathbb{R}^d$ is a fixed vector satisfying $\|r\|_2 = \sqrt{d}$; for instance, $r = (1, 1, ..., 1)^T$. The interpolant $I_t = \alpha_t z + \beta_t x_1$ where $z \sim \mathsf{N}(0, \mathrm{I}) \perp x_1 \sim \mu^*$.*

*Using the general formula in Appendix C.2, we get $b_t(x) = \dot{\beta}_t r\tanh(h + \beta_t\langle r, x\rangle)$. Then $\nabla b_t(x) = \dot{\beta}_t\beta_t rr^T \mathrm{sech}^2(h + \beta_t\langle r, x\rangle)$, which yields*

$$
\|\nabla b_t(x)\|_2^2 = d\dot{\beta}_t^2\beta_t^2 \mathrm{sech}^4(h + \beta_t\langle r, x\rangle)\,.
$$

*This is effectively the same as the 1D example in Proposition 3.6. Using the result there, we get that the optimal $\beta_t, \alpha_t = \sqrt{1 - \beta_t^2}$ minimizing $A_2$ satisfies*

$$
t = \frac{\int_0^{\beta_t} u(G(u))^{1/2}\mathrm{d}u}{\int_0^1 u(G(u))^{1/2}\mathrm{d}u}\,.
$$

*where $G(u) = \mathbb{E}[\mathrm{sech}^4(h + u\langle r, \sqrt{1 - u^2}z + ux_1\rangle)]$. Again, an approximate solution is*

$$
\beta_t = \frac{1}{\sqrt{d}}\sqrt{-\log(1 + (e^{-d} - 1)t)}\,. \tag{3.14}
$$

Beyond the above examples, we have a general formula for optimizing $A_2$ over scalar interpolation schedules, for general distributions.

**Example 3.13** (Optimizing avg-Lip$^2$ for general distributions)**.** *Consider a general distribution $\mu^*$ in $d$ dimensions and we assume it to be smooth for simplicity. Let $b^\dagger(x)$ be defined as in Proposition 3.1 and let $\alpha_t = \sqrt{1 - \beta_t^2}$. Then using Proposition 3.1,*

$$
b_t(x) = \dot{\beta}_t\left(\frac{-\beta_t}{1 - \beta_t^2}x + \frac{1}{1 - \beta_t^2}\left((1 - t^\dagger)b_{t^\dagger}^\dagger(\frac{t^\dagger}{\beta_t}x) + x\right)\right)\,,
$$

*and*

$$\nabla b_t(x) = \dot{\beta}_t \left( \frac{-\beta_t}{1-\beta_t^2}\mathrm{I} + \frac{1}{1-\beta_t^2}\left( (1-t^\dagger)\frac{t^\dagger}{\beta_t}\nabla b_{t^\dagger}^\dagger(\frac{t^\dagger}{\beta_t}x) + \mathrm{I} \right) \right) = \dot{\beta}_t F(\beta_t, x),$$

*where we denote the term in the big bracket by $F(\beta_t, x)$. Then*

$$A_2 = \int_0^1 \mathbb{E}[\|\nabla b_t(I_t)\|_2^2] = \int_0^1 \dot{\beta}_t^2 \mathbb{E}[\|F(\beta_t, I_t)\|_2^2]\mathrm{d}t = \int_0^1 \dot{\beta}_t^2 G(\beta_t)\mathrm{d}t,$$

*where we denote $G(\beta_t) = \mathbb{E}[\|F(\beta_t, I_t)\|_2^2]$. Solving the Euler-Lagrange equation with the Beltrami Identity (see Appendix C.3) leads to the equation that $\beta_t$ satisfies:*

$$t = \frac{\int_0^{\beta_t}(G(u))^{1/2}\mathrm{d}u}{\int_0^1(G(u))^{1/2}\mathrm{d}u}.$$

*In general, finding the optimal $\beta_t$ analytically is challenging. While numerical solutions are possible once $b^\dagger$ is available, it is computationally costly in high dimensions as we need to evaluate $G$. Our previous examples demonstrate that certain cases allow for simpler solutions. In particular, we have an analytic formula for the Gaussian case. For Gaussian mixture distributions, we can derive approximate analytical solutions, and for log-concave cases, we can leverage insights from the Gaussian analysis to construct schedules that achieve our numerical objectives.*

## 4 Numerical Demonstrations

We now turn to numerical experiments. We consider four target distributions: high-dimensional Gaussian random fields, high-dimensional Gaussian mixtures, and the invariant measures of the stochastic Allen–Cahn and Navier–Stokes equations. These range from exactly Gaussian to strongly non-Gaussian, and the goal is to test how well the schedule design developed analytically in Section 3 carries over to settings of practical interest.

In all experiments, the generative process is the ODE associated with the linear stochastic interpolant; the designed schedule is applied at inference time via the transfer formula (Proposition 3.1), so the same trained drift is reused across schedules. For Gaussian and Gaussian-mixture targets, the drift is available in closed form and we evaluate it directly; for the Allen–Cahn and Navier–Stokes targets, the drift is approximated by a UNet Ho et al. (2020) trained with the linear schedule, with training procedure and hyperparameters following Chen et al. (2024c). For numerical stability, the ODE is integrated from $t_{\min} = 10^{-3}$ to $t_{\max} = 1 - 10^{-3}$ throughout. Code to reproduce all results will be made publicly available upon publication[5]; experimental details for the Navier–Stokes case are collected in Appendix D.

Sampling accuracy is assessed via spectral diagnostics. For a generated sample $u$ (one- or two-dimensional in this paper), we compute the radially binned energy spectrum

$$E(k) = \sum_{k \le |m|_2 < k+1} |\hat{u}(m)|^2,$$

where $\hat{u}(m)$ are the Fourier coefficients; in the Navier–Stokes case with vorticity formulation, this is the enstrophy spectrum. Spectra are averaged over an ensemble of independently generated samples.

### 4.1 Gaussians

We first consider a Gaussian random field with distribution $\mathsf{N}(0, \sigma^2(-\Delta + \tau^2\mathrm{I})^{-s})$ on $D = [0,1]^2$, where $-\Delta$ denotes the Laplacian with homogeneous Dirichlet boundary conditions. We fix $s = 3$, $\tau = 1$, and $\sigma^2 = (4\pi^2 + \tau^2)^s$, so that the field exhibits significant scale separation across Fourier modes. Target samples $x_1$ are drawn exactly from this distribution.

---

[5]Anonymized for double-blind review; the repository will be linked in the camera-ready version.

The noise variable $z$ in the stochastic interpolant is sampled from spatial white noise, corresponding to the Gaussian random field with $s = 0$ and unit variance. The field is discretized on a uniform $N \times N$ grid, and the resulting generative ODE is integrated using a fixed-step fourth-order Runge–Kutta (RK4) method.

We compare the standard linear schedule $\beta_t = t$ with the designed schedule (3.11), which minimizes avg-Lip$^2$ for Gaussian targets. Figure 2 shows representative samples at resolution $N = 128$ generated using 20 RK4 steps. The designed schedule produces visibly smoother and more coherent samples. The right panel illustrates the schedules themselves, highlighting the rapid initial growth of the designed schedule, which reflects the fast speed required at early times.

Figure 3 compares the energy spectra of the true distribution and generated samples across resolutions. The designed schedule yields significantly more accurate spectra and maintains accuracy as $N$ increases (due to the logarithmic scaling), whereas the linear schedule deteriorates under refinement, reflecting its poorer numerical conditioning.

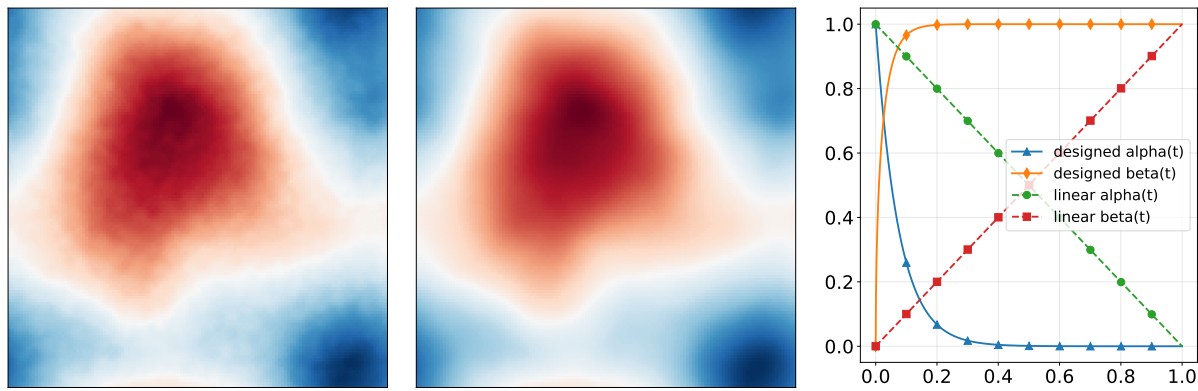

Figure 2: Left: $128 \times 128$ Gaussian fields generated by using linear schedules with 20 steps of the RK4 integrator. Middle: $128 \times 128$ Gaussian fields generated by using the designed schedules with 20 steps of the RK4 integrator. Right: linear and designed schedules.

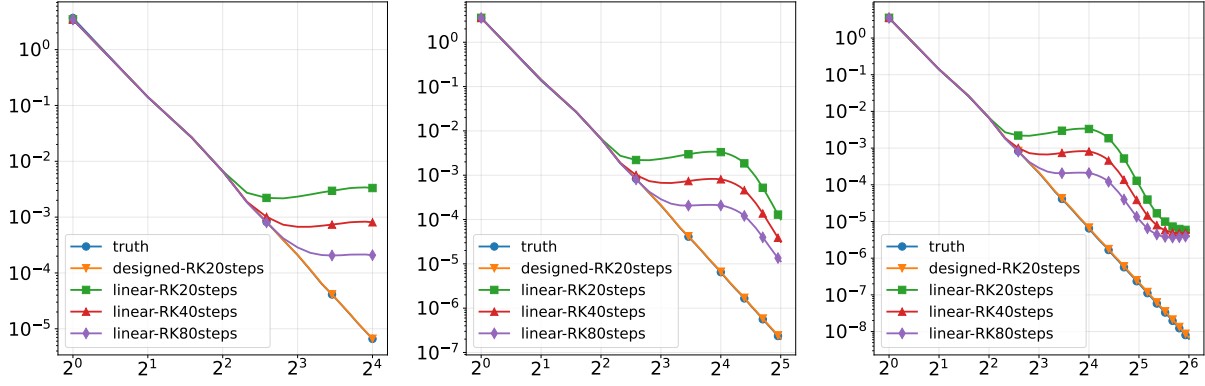

Figure 3: Energy spectra of Gaussian fields: comparison between truth, generated via designed schedules or standard linear schedules, with $20, 40$ or $80$ RK4 steps. The three figures correspond to different resolutions. Left: $32 \times 32$; middle: $64 \times 64$; right: $128 \times 128$.

## 4.2 Gaussian mixtures

We next consider the high-dimensional Gaussian mixture distribution defined in (3.13) with dimension $d = 1000$, mixture weight $p = 0.3$, and mean vector $r = (1, 1, \ldots, 1) \in \mathbb{R}^d$. This distribution is strongly non-Gaussian and exhibits well-separated modes. The noise variable $z$ is sampled from $\mathsf{N}(0, \mathsf{I})$.

We compare the linear schedule $\beta_t = t$ with the approximate min-avg-Lip$^2$ schedule (3.14), taking $\alpha_t = \sqrt{1 - \beta_t^2}$ in both cases. The drift field is given analytically by Example 3.12, allowing us to isolate the numerical effects of the schedule without learning error. The ODE is integrated using only 2, 3, or 4 RK4 steps, with $10^4$ independent samples generated for each setting.

To assess mode recovery, we project the generated samples onto one dimension using PCA and fit a one-dimensional bimodal Gaussian mixture model. Table 1 reports the estimated smaller mixture weight. The linear schedule frequently collapses to a single mode when using few steps, while the approximate min-avg-Lip$^2$ schedule accurately recovers both modes even under extremely coarse discretization.

|  | Truth | Linear schedule | Approx min-avg-Lip$^2$ schedule |
|---|---|---|---|
| 2 RK4 steps | 0.3 | 0.00 | 0.42 |
| 3 RK4 steps | 0.3 | 0.03 | 0.26 |
| 4 RK4 steps | 0.3 | 0.09 | 0.27 |

Table 1: True and estimated weights of one mode recovered from the samples (values reported to 2 decimal places). We obtain two weights since we fit a bimodal GMM, and we always report the smaller weight.

## 4.3 Invariant distributions of stochastic Allen-Cahn

We consider the invariant distribution of the stochastic Allen–Cahn equation on $[0, 1]$, formally given by

$$\exp\left(-\int_0^1 \frac{1}{2}(\partial_x u(x))^2 + V(u(x))\,\mathrm{d}x\right), \qquad V(u) = (1 - u^2)^2.$$

This distribution is bimodal and moderately non-Gaussian, with samples concentrating near $u = \pm 1$. We discretize the equation using finite differences on $N$ equispaced grid points, yielding an $N$-dimensional target distribution.

Samples $x_1$ from the invariant distribution are generated using ensemble MCMC methods Chen (2025), while $z$ is sampled from spatial white noise. The drift is trained under the linear schedule, and the designed schedule is applied at inference time via the transfer formula (Proposition 3.1); we then compare the resulting energy spectra.

The designed schedule is obtained by treating the Gaussian reference measure

$$\exp\left(-\int_0^1 \tfrac{1}{2}(\partial_x u)^2 \,\mathrm{d}x\right)$$

as a proxy for the covariance structure and applying the avg-Lip$^2$-optimal schedule of Proposition 3.9. Figure 4 shows that this schedule yields consistently more accurate energy spectra than the linear schedule, and that the improvement is robust across spatial resolutions. With only 10 RK4 steps, the designed schedule produces consistent fine-scale spectra across resolutions $N = 32, 64, 128$, in line with the logarithmic dependence of the drift Lipschitz constant on resolution; the linear schedule, by contrast, exhibits a polynomial growth of stiffness and requires substantially more steps to reach comparable accuracy.

## 4.4 Invariant distributions of stochastic Navier-Stokes

Finally, we consider invariant distributions of the two-dimensional stochastically forced Navier–Stokes equations on the torus $\mathbb{T}^2 = [0, 2\pi]^2$. Using the vorticity formulation, the dynamics are given by

$$\mathrm{d}\omega + v \cdot \nabla\omega \, \mathrm{d}t = \nu\Delta\omega \, \mathrm{d}t - \alpha\omega \, \mathrm{d}t + \varepsilon \, \mathrm{d}\eta, \tag{4.1}$$

where $v = \nabla^\perp\psi = (-\partial_y\psi, \partial_x\psi)$ is the incompressible velocity field associated with the stream function $\psi$ satisfying $-\Delta\psi = \omega$. We fix the parameters to $\nu = 10^{-3}$, $\alpha = 0.1$, and $\varepsilon = 1$. The stochastic forcing $\eta$ is white in time and acts on a finite number of low-frequency Fourier modes, following the setup in Chen

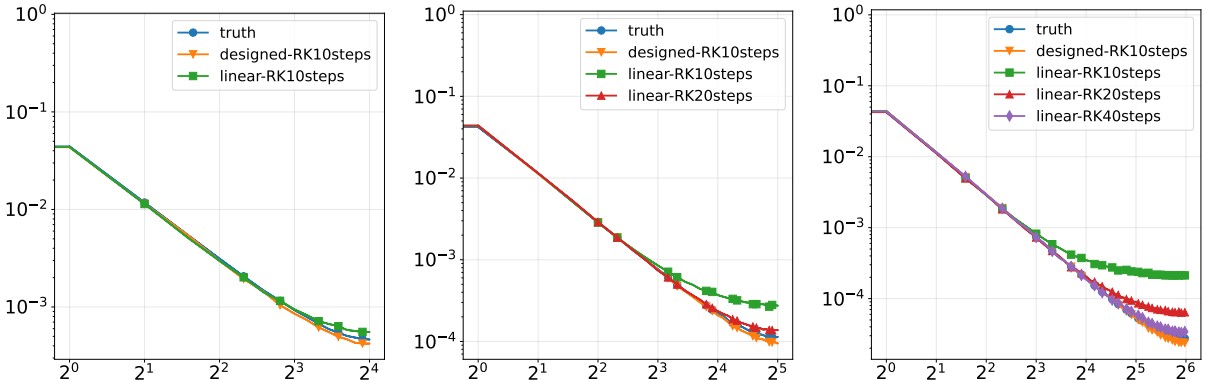

Figure 4: Energy spectra of invariant distributions of stochastic Allen-Cahn: comparison between truth, generated via designed schedules or standard linear schedules, with $10, 20$ or $40$ RK4 steps. The three figures correspond to different resolutions. Left: $32$; middle: $64$; right: $128$.

et al. (2024c). Under these choices, the system is ergodic and admits a unique invariant probability measure Hairer & Mattingly (2006).

Samples from the invariant distribution are generated via long-time numerical simulation of (4.1) using a pseudo-spectral method with standard de-aliasing. After an initial burn-in period sufficient for equilibration, vorticity snapshots are collected at regular time intervals to form an approximately independent dataset of samples. In the stochastic interpolant framework, samples $x_1$ are drawn from this empirical invariant distribution, while the initial condition $z$ is sampled from a spatial Gaussian random field. We train an ODE-based generative model to transport $z$ to $x_1$ over unit time using a drift field parameterized by a UNet architecture; full architectural and training details are reported in Appendix D.

To probe the effect of interpolation schedules, we compare the linear schedule $\beta_t = t$ with the designed schedule (3.11). The schedule parameter $\lambda^\star$ in Proposition 3.9 is the smallest eigenvalue of the target covariance, but for the highly non-Gaussian invariant measure of (4.1) it is not directly accessible. We therefore use a data-driven proxy: at the finest resolved frequency $k_{\max}$,

$$\lambda^\star = \frac{S_{\mathrm{data}}(k_{\max})}{S_{\mathrm{noise}}(k_{\max})}, \tag{4.2}$$

where $S_{\mathrm{data}}, S_{\mathrm{noise}}$ denote the radially-averaged enstrophy spectra of the data and the noise, respectively. This rule encodes the same intuition as the eigenvalue ratio in the Gaussian case: it is the worst-case ratio of target to source variance among the resolved Fourier modes. Empirically it gives $\lambda^\star \approx 3 \times 10^{-4}$ at $64 \times 64$ and $\lambda^\star \approx 10^{-5}$ at $128 \times 128$. The designed schedule is applied at inference time via the transfer formula (Proposition 3.1), so both schedules use the same trained drift; the ODE is integrated with a fixed-step RK4 method on a uniform grid in $[t_{\min}, t_{\max}]$.

Figure 5 illustrates representative generated samples and ensemble-averaged enstrophy spectra at $128 \times 128$ with 10 RK4 steps. The designed schedule yields visibly smoother samples without spurious fine-scale artifacts and reproduces the enstrophy spectrum across all frequencies, while the linear schedule overestimates fine-scale energy by orders of magnitude and requires more than 20 steps to reach a similar level of accuracy.

To quantify accuracy beyond the global spectrum, we report relative enstrophy errors decomposed into mid $(8 \leq k < 24)$ and high $(k \geq 24)$ wavenumber bands. For a band $B$, the relative error is $\frac{1}{|B|} \sum_{k \in B} |S_{\mathrm{gen}}(k) - S_{\mathrm{truth}}(k)|/|S_{\mathrm{truth}}(k)|$.

Figure 6 and Table 2 report band errors as a function of RK4 step count for both $64 \times 64$ and $128 \times 128$. The designed schedule consistently dominates the linear schedule. The improvement is most dramatic at high wavenumbers: at $128 \times 128$ with 10 RK4 steps, the linear schedule yields a relative error above $350\%$ in the high band, while the designed schedule reaches $16\%$ – a $20\times$ reduction. The mid-band error of the

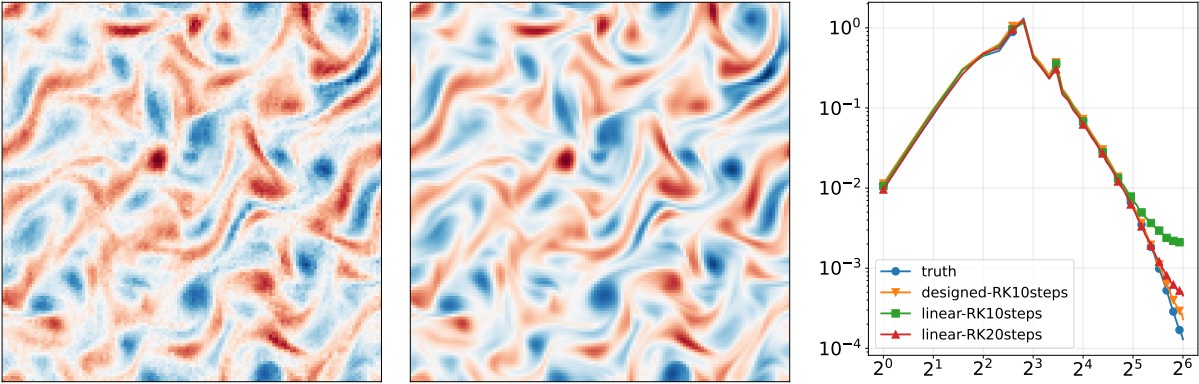

Figure 5: Comparison of generated $128 \times 128$ vorticity fields using 10 RK4 steps. Left: linear schedule; middle: designed schedule; right: enstrophy spectra of 500 samples generated with each schedule, against the truth.

designed schedule is also roughly half of the linear schedule's at every step count, and barely changes with more RK4 steps – already at 10 steps it has converged. In contrast, the linear schedule needs 50 RK4 steps simply to bring the high-band error below 30% at $128 \times 128$. The point is that, with the linear schedule, the large-$k$ modes are integrated with insufficient resolution and remain inaccurate, while the designed schedule allocates time appropriately and resolves them at small step counts.

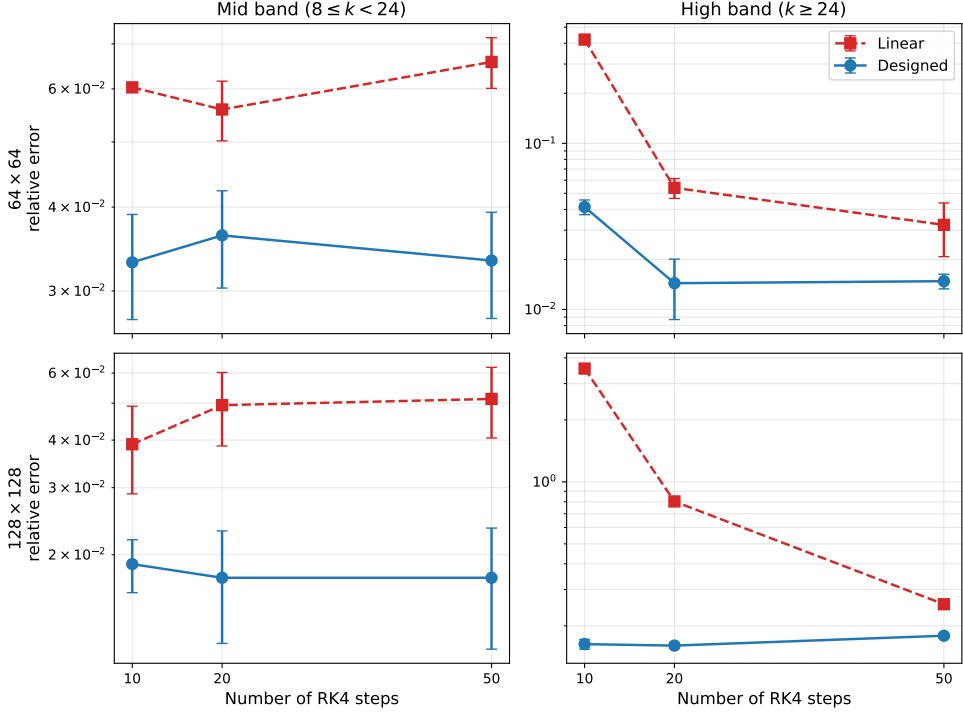

Figure 6: Per-band relative enstrophy error vs. number of RK4 steps for the linear schedule (red, dashed) and the designed schedule (blue, solid). Top row: $64 \times 64$; bottom row: $128 \times 128$. Left column: mid band ($8 \leq k < 24$); right column: high band ($k \geq 24$). Mean and standard deviation over three random seeds, 500 samples per seed. The designed schedule converges already at 10 steps; the linear schedule needs many more steps to resolve the large-$k$ modes.

| Resolution | Method (steps) | Mid ($8 \leq k < 24$) | High ($k \geq 24$) |
|---|---|---|---|
| $64 \times 64$ | Linear (10) | $0.060 \pm 0.001$ | $0.421 \pm 0.011$ |
| $64 \times 64$ | Linear (20) | $0.056 \pm 0.006$ | $0.054 \pm 0.007$ |
| $64 \times 64$ | Linear (50) | $0.066 \pm 0.006$ | $0.032 \pm 0.012$ |
| $64 \times 64$ | Designed (10) | $\mathbf{0.033 \pm 0.006}$ | $\mathbf{0.041 \pm 0.004}$ |
| $64 \times 64$ | Designed (20) | $\mathbf{0.036 \pm 0.006}$ | $\mathbf{0.014 \pm 0.006}$ |
| $64 \times 64$ | Designed (50) | $\mathbf{0.033 \pm 0.006}$ | $\mathbf{0.015 \pm 0.002}$ |
| $128 \times 128$ | Linear (10) | $0.039 \pm 0.010$ | $3.549 \pm 0.049$ |
| $128 \times 128$ | Linear (20) | $0.049 \pm 0.011$ | $0.803 \pm 0.013$ |
| $128 \times 128$ | Linear (50) | $0.051 \pm 0.011$ | $0.255 \pm 0.004$ |
| $128 \times 128$ | Designed (10) | $\mathbf{0.019 \pm 0.003}$ | $\mathbf{0.163 \pm 0.009}$ |
| $128 \times 128$ | Designed (20) | $\mathbf{0.017 \pm 0.006}$ | $\mathbf{0.160 \pm 0.005}$ |
| $128 \times 128$ | Designed (50) | $\mathbf{0.017 \pm 0.006}$ | $\mathbf{0.179 \pm 0.005}$ |

Table 2: Relative enstrophy spectrum error in the mid and high wavenumber bands for the linear and designed schedules at resolutions $64 \times 64$ and $128 \times 128$. Mean and standard deviation reported over three independent random seeds, with 500 generated samples per seed. Bold indicates the lower mean per (resolution, step count, band). Low-band ($k < 8$) errors are dominated by model approximation and are similar across schedules; we omit them here.

**Non-Gaussian metrics.** Spectra capture only second-order statistics, while the invariant distribution of (4.1) is strongly non-Gaussian. To probe this, we evaluate the flatness $F(r) = S_4(r)/S_2(r)^2$ of vorticity increments at scales $r \in \{1, 2\}$ pixels (with $S_p(r) = \mathbb{E}|\omega(\cdot + r) - \omega(\cdot)|^p$), the gradient kurtosis (equivalently $F(1)$), and the Kolmogorov–Smirnov distance between the empirical pixel distributions of generated and ground-truth samples. A Gaussian field has flatness 3; departures from 3 quantify intermittency. At $128 \times 128$ (Table 3), the designed schedule recovers the truth flatness to within 3% at 10 RK4 steps, whereas the linear schedule underestimates flatness by 13% and reaches comparable accuracy only after 50 steps. The KS distance is also smaller for the designed schedule. Despite the strongly non-Gaussian nature of the invariant measure, schedules optimized based on Gaussian analysis still provide significant improvements in fine-scale and intermittent statistics.

| Method (steps) | Flatness $r=1$ | Flatness $r=2$ | Gradient kurtosis | KS distance ($\times 10^{-3}$) |
|---|---|---|---|---|
| Truth | 4.95 | 4.34 | 4.95 | — |
| Linear (10) | $4.29 \pm 0.02$ | $4.13 \pm 0.02$ | $4.29 \pm 0.02$ | 4.87 |
| Linear (20) | $4.67 \pm 0.03$ | $4.27 \pm 0.03$ | $4.67 \pm 0.03$ | 5.33 |
| Linear (50) | $4.69 \pm 0.03$ | $4.28 \pm 0.03$ | $4.69 \pm 0.03$ | 5.24 |
| Designed (10) | $\mathbf{4.82 \pm 0.03}$ | $\mathbf{4.31 \pm 0.03}$ | $\mathbf{4.82 \pm 0.03}$ | $\mathbf{4.71}$ |
| Designed (20) | $\mathbf{4.83 \pm 0.03}$ | $\mathbf{4.32 \pm 0.03}$ | $\mathbf{4.83 \pm 0.03}$ | $\mathbf{4.58}$ |
| Designed (50) | $\mathbf{4.83 \pm 0.03}$ | $\mathbf{4.32 \pm 0.03}$ | $\mathbf{4.83 \pm 0.03}$ | $\mathbf{4.61}$ |

Table 3: Non-Gaussian statistics of generated $128 \times 128$ vorticity fields. Flatness $F(r) = S_4(r)/S_2(r)^2$ measures intermittency at increment scale $r$ (Gaussian baseline 3); the gradient kurtosis equals $F(1)$. KS distance compares pixel-value distributions on $10^5$ random pixels. Mean and standard deviation reported over five seeds, 500 samples per seed. Truth values evaluated on the test set.

## 5 Conclusions

We have studied the design of interpolation schedules in flow and diffusion-based generative models within the stochastic interpolants framework, from a combined statistical and numerical perspective.

On the statistical side, we showed that all scalar interpolation schedules are equivalent under the Kullback–Leibler divergence in path space, once the diffusion coefficient is tuned a posteriori. This indicates that, within the scalar class, schedule choice is not a purely statistical question and the criterion for selection must come from elsewhere.

On the numerical side, we proposed minimizing the averaged squared Lipschitzness of the drift, in contrast with kinetic-energy minimization in optimal transport. A simple transfer formula expresses the drift of one scalar schedule in terms of the drift of another, so the designed schedule can be deployed at inference time on a model trained under a different (e.g., linear) schedule, without retraining. For Gaussian targets the designed schedule gives an exponential reduction in the Lipschitz constant of the drift; for Gaussian-mixture targets it mitigates few-step mode collapse. These analytical findings carry over to invariant measures of stochastic Allen–Cahn and Navier–Stokes equations, where the designed schedule yields more accurate fine-scale spectra at fixed integrator budget.

A natural direction for future work is to go beyond scalar schedules. Matrix-valued, nonlinear, or instance-dependent schedules may break the statistical equivalence we established and offer further numerical regularization, although their analysis and parameterization raise additional questions. Combining schedule design with higher-order or multiscale integrators, and with consistency or flow-map distillation, are other promising avenues.

**Broader Impact Statement**

This work is methodological and analyzes the design of interpolation schedules in flow- and diffusion-based generative models. We do not foresee direct negative societal impacts beyond those generally associated with generative modeling research.

**Acknowledgments**

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

# A    Sketch of Derivations for Stochastic Interpolants

*Sketch of derivation for Proposition 2.2.* For any smooth test function $\phi : \mathbb{R}^d \to \mathbb{R}$,

$$\mathrm{d}\phi(I_t) = \dot{I}_t \cdot \nabla\phi(I_t)\mathrm{d}t\,. \tag{A.1}$$

We denote by $\mu(t, \mathrm{d}x)$ the measure of $I_t$. Then,

$$\int_{\mathbb{R}^d} \phi(x)\mu(t, \mathrm{d}x) = \mathbb{E}[\phi(I_t)] = \mathbb{E}[\phi(I_0)] + \int_0^t \mathbb{E}[\dot{I}_s \cdot \nabla\phi(I_s)]\mathrm{d}s\,. \tag{A.2}$$

Using the definition of conditional expectation, we have the identity

$$\mathbb{E}[\dot{I}_s \cdot \nabla\phi(I_s)] = \mathbb{E}[\mathbb{E}[\dot{I}_s|I_s] \cdot \nabla\phi(I_s)] = \int_{\mathbb{R}^d} \mathbb{E}[\dot{I}_s|I_s = x] \cdot \nabla\phi(x)\mu(s, \mathrm{d}x)\,. \tag{A.3}$$

Combining the above two equations lead to

$$\int_{\mathbb{R}^d} \phi(x)\mu(t, \mathrm{d}x) = \int_{\mathbb{R}^d} \phi(x)\mu(0, \mathrm{d}x) + \int_0^t \int_{\mathbb{R}^d} \mathbb{E}[\dot{I}_s|I_s = x] \cdot \nabla\phi(x)\mu(s, \mathrm{d}x)\mathrm{d}s\,, \tag{A.4}$$

which implies $\mu(t, \cdot)$ is the weak solution to the transport equation corresponding to the ODE $\mathrm{d}X_t = b_t(X_t)\mathrm{d}t$ with $b_t(x) = \mathbb{E}[\dot{I}_t|I_t = x]$. $\qquad\square$

*Sketch of derivation for Proposition 2.3.* Assume the density of $I_t$ exists and denote it by $\rho_t$. By Proposition 2.2, $\rho_t$ satisfies the transport equation

$$\partial_t\rho_t + \nabla \cdot (\rho_t b_t) = 0\,.$$

Using the fact that $\nabla \cdot (\rho\nabla\log\rho) = \Delta\rho$, we can rewrite the equation as

$$\partial_t\rho_t + \nabla \cdot (\rho_t(b_t + \epsilon_t\nabla\log\rho_t)) = \epsilon_t\Delta\rho_t\,,$$

which is exactly the Fokker-Planck equation corresponding to the SDE

$$\mathrm{d}X_t = (b_t(X_t) + \epsilon_t\nabla\log\rho_t(X_t))\,\mathrm{d}t + \sqrt{2\epsilon_t}\mathrm{d}W_t\,.$$

$\qquad\square$

*Sketch of derivation for* (2.2). The second equation in (2.2) follows directly from the first one. Here we derive the first one. Let us denote the density of $\beta_t x_1$ by $q_t$. Then $I_t$ is a Gaussian noisy version of $\beta_t x_1$, implying that

$$\rho_t(x) \propto \int_{\mathbb{R}^d} q_t(y)\exp(-\frac{\|x - y\|_2^2}{2\alpha_t^2})\mathrm{d}y\,.$$

Taking gradient yields the formula

$$\nabla \log \rho_t(x) = \frac{1}{\int_{\mathbb{R}^d} q_t(y) \exp(-\frac{\|x-y\|_2^2}{2\alpha_t^2})\mathrm{d}y} \int_{\mathbb{R}^d} (-\frac{x-y}{\alpha_t^2})q_t(y)\exp(-\frac{\|x-y\|_2^2}{2\alpha_t^2})\mathrm{d}y \,.$$

On the other hand, by the Bayes rule, we know that

$$\frac{1}{\int_{\mathbb{R}^d} q_t(y)\exp(-\frac{\|x-y\|_2^2}{2\alpha_t^2})\mathrm{d}y} q_t(y)\exp(-\frac{\|x-y\|_2^2}{2\alpha_t^2})$$

is the density of the conditional distribution $\beta_t x_1 | \alpha_t z + \beta_t x_1 = x$. Therefore,

$$\nabla \log \rho_t(x) = \mathbb{E}[-\frac{x-\beta_t x_1}{\alpha_t^2}|I_t = x] = -\mathbb{E}[\frac{z}{\alpha_t}|I_t = x]\,.$$

This leads to the first formula in (2.2). $\qquad\square$

## B Discussion on SDEs with Singular Drift

In Section 2.3, the optimal diffusion coefficient is

$$\epsilon_t = \alpha_t^2(\frac{\dot{\beta}_t}{\beta_t} - \frac{\dot{\alpha}_t}{\alpha_t})\,.$$

With this choice and using the identities in (2.2), we obtain the following SDE

$$\mathrm{d}X_t = (2b_t(X_t) - \frac{\dot{\beta}_t}{\beta_t}X_t)\mathrm{d}t + \sqrt{2\epsilon_t}\mathrm{d}W_t\,.$$

For example, we take $\beta_t = t, \alpha_t = 1-t$, which yields

$$\mathrm{d}X_t = (2b_t(X_t) - \frac{1}{t}X_t)\mathrm{d}t + \sqrt{2\frac{1-t}{t}}\mathrm{d}W_t\,.$$

The diffusion coefficient is singular and appears worrisome. However, note that

$$\mathrm{d}(tX_t) = 2tb_t(X_t)\mathrm{d}t + \sqrt{2t(1-t)}\mathrm{d}W_t\,,$$

which implies that

$$X_t = \frac{1}{t}\int_0^t 2sb_s(X_s)\mathrm{d}s + \frac{1}{t}\int_0^t \sqrt{2s(1-s)}\mathrm{d}W_s\,.$$

The last term is well defined as

$$\frac{1}{t^2}\int_0^t 2s(1-s)\mathrm{d}s = 1 - \frac{2}{3}t$$

is non-singular as $t \to 0$. Therefore, the above stochastic integral equation is well defined. One can use Picard's iteration to prove the existence of a solution rigorously.

## C Technical Details for Optimizing Averaged Squared Lipschitzness

### C.1 Optimal transport drift in the 1D Gaussian case

We provide a sketch of proof for claims made in Remark 3.4. In the Gaussian setting, optimal transport theory implies that the optimal transport map satisfies $Tx = C_0^{-\frac{1}{2}}(C_0^{\frac{1}{2}}MC_0^{\frac{1}{2}})^{\frac{1}{2}}C_0^{-\frac{1}{2}}x = \sqrt{\frac{M}{C_0}}x$ in 1D. Therefore, the variance at time $t$ in the optimal transport path satisfies

$$C_t = ((1-t)I + tT)C_0((1-t)I + tT)^T\,.$$

Differentiation over $t$ leads to

$$\dot{C}_t = (T - I)C_0((1 - t)I + tT)^T + ((1 - t)I + tT)C_0(T - I)^T .$$

On the other hand, let $b_t(x) = A_t x$, then using the ODE $\dot{x}_t = A_t x_t$ and differentiating $C_t = \mathbb{E}[x_t x_t^T]$ leads to the equation $\dot{C}_t = A_t C_t + C_t A_t^T$. Comparing the above two formulas for $\dot{C}_t$ implies

$$A_t = (T - I)((1 - t)I + tT)^{-1} .$$

For 1D, we obtain the formula

$$b_t(x) = \frac{\sqrt{M} - \sqrt{C_0}}{(1 - t)\sqrt{C_0} + t\sqrt{M}} x .$$

In particular, we take $C_0 = 1$ to get

$$b_t(x) = \frac{\sqrt{M} - 1}{1 - t + t\sqrt{M}} x .$$

### C.2 Formula for Gaussian mixtures

We provide exact formula for the Gaussian mixture model (GMM).

**Proposition C.1.** *Let the target density be a GMM with $J \in \mathbb{N}$ modes*

$$\rho^\star(x) = \sum_{j=1}^{J} p_j \mathsf{N}(x; m_j, C_j) \tag{C.1}$$

*where $p_j \geq 0$ with $\sum_{j=1}^{J} p_j = 1$, $m_j \in \mathbb{R}^d$, and $C_j = C_j^T \in \mathbb{R}^d \times \mathbb{R}^d$ positive-definite. Then*

$$b_t(x) = \dot{\beta}_t \frac{\sum_{j=1}^{J} p_j m_j \mathsf{N}(x; \overline{m}_j(t), \overline{C}_j(t))}{\sum_{j=1}^{J} p_j \mathsf{N}(x; \overline{m}_j(t), \overline{C}_j(t))}$$
$$+ \frac{\sum_{j=1}^{J} p_j (\beta_t \dot{\beta}_t C_j + \alpha_t \dot{\alpha}_t I)\overline{C}_j^{-1}(t)(x - \overline{m}_j(t))\mathsf{N}(x; \overline{m}_j(t), \overline{C}_j(t))}{\sum_{j=1}^{J} p_j \mathsf{N}(x; \overline{m}_j(t), \overline{C}_j(t))} \tag{C.2}$$

*where*

$$\overline{m}_j(t) = \beta_t m_j, \qquad \overline{C}_j(t) = \beta_t^2 C_j + \alpha_t^2 I . \tag{C.3}$$

*Proof.* By definition

$$\begin{aligned} b_t(x) &= \mathbb{E}[\dot{\beta}_t x_1 + \dot{\alpha}_t z | I_t = x] \\ &= \mathbb{E}[\dot{\beta}_t \beta_t^{-1}(x - \alpha_t z) + \dot{\alpha}_t z | I_t = x] \\ &= \dot{\beta}_t \beta_t^{-1} x + \alpha_t(\alpha_t \dot{\beta}_t \beta_t^{-1} - \dot{\alpha}_t)\nabla \log \rho_t(x) . \end{aligned} \tag{C.4}$$

where we used the fact $\nabla \log \rho_t(x) = -\alpha_t^{-1}\mathbb{E}[z | I_t = x]$. For the GMM,

$$\rho_t(x) = \sum_{j=1}^{J} p_j \mathsf{N}(x; \overline{m}_j(t), \overline{C}_j(t)) , \tag{C.5}$$

so that

$$\nabla \log \rho_t(x) = -\frac{\sum_{j=1}^{J} p_j \overline{C}_j^{-1}(t)(x - \overline{m}_j(t))\mathsf{N}(x; \overline{m}_j(t), \overline{C}_j(t))}{\sum_{j=1}^{J} p_j \mathsf{N}(x; \overline{m}_j(t), \overline{C}_j(t))} . \tag{C.6}$$

Inserting this expression in (C.4) we obtain

$$
\begin{aligned}
&\frac{\dot{\beta}_t}{\beta_t} x + \alpha_t^2 \frac{\dot{\beta}_t}{\beta_t} \nabla \log \rho_t(x) \\
&= \frac{\dot{\beta}_t}{\beta_t} \left( x - \frac{\sum_{j=1}^J p_j (I - \beta_t^2 C_j \overline{C}_j^{-1}(t))(x - \overline{m}_j(t)) \mathsf{N}(x; \overline{m}_j(t), \overline{C}_j(t))}{\sum_{j=1}^J p_j \mathsf{N}(x; \overline{m}_j(t), \overline{C}_j(t))} \right) \\
&= \frac{\dot{\beta}_t}{\beta_t} \left( \frac{\sum_{j=1}^J p_j \big(\beta_t m_j + \beta_t^2 C_j \overline{C}_j^{-1}(x - \overline{m}_j)\big) \mathsf{N}(x; \overline{m}_j(t), \overline{C}_j(t))}{\sum_{j=1}^J p_j \mathsf{N}(x; \overline{m}_j(t), \overline{C}_j(t))} \right) \\
&= \dot{\beta}_t \frac{\sum_{j=1}^J p_j m_j \mathsf{N}(x; \overline{m}_j(t), \overline{C}_j(t))}{\sum_{j=1}^J p_j \mathsf{N}(x; \overline{m}_j(t), \overline{C}_j(t))} + \frac{\sum_{j=1}^J p_j \beta_t \dot{\beta}_t C_j \overline{C}_j^{-1}(x - \bar{m}_j) \mathsf{N}(x; \overline{m}_j(t), \overline{C}_j(t))}{\sum_{j=1}^J p_j \mathsf{N}(x; \overline{m}_j(t), \overline{C}_j(t))} \,,
\end{aligned}
\tag{C.7}
$$

where in the first and second identities, we used the fact that $\alpha_t^2 \overline{C}_j^{-1}(t) = I - \beta_t^2 C_j \overline{C}_j^{-1}(t)$.

Now, using $b_t(x) = \dot{\beta}_t \beta_t^{-1} x + \alpha_t^2 (\dot{\beta}_t \beta_t^{-1} - \dot{\alpha}_t) \nabla \log \rho_t(x)$, we get the final formula. $\qquad \square$

*Remark* C.2. This form of the formula holds generally when $z$ is not of unit covariance. Let $z \sim \mathsf{N}(0, C_0)$, then we have

$$
\begin{aligned}
b_t(x) = \dot{\beta}_t &\frac{\sum_{j=1}^J p_j m_j \mathsf{N}(x; \overline{m}_j(t), \overline{C}_j(t))}{\sum_{j=1}^J p_j \mathsf{N}(x; \overline{m}_j(t), \overline{C}_j(t))} \\
&+ \frac{\sum_{j=1}^J p_j (\beta_t \dot{\beta}_t C_j + \alpha_t \dot{\alpha}_t C_0) \overline{C}_j^{-1}(t)(x - \overline{m}_j(t)) \mathsf{N}(x; \overline{m}_j(t), \overline{C}_j(t))}{\sum_{j=1}^J p_j \mathsf{N}(x; \overline{m}_j(t), \overline{C}_j(t))}
\end{aligned}
\tag{C.8}
$$

where

$$
\overline{m}_j(t) = \beta_t m_j, \qquad \overline{C}_j(t) = \beta_t^2 C_j + \alpha_t^2 C_0 \,.
\tag{C.9}
$$

When there is only one mode, we get

$$
b_t(x) = \dot{\beta}_t m_1 + (\alpha_t \dot{\alpha}_t C_0 + \beta_t \dot{\beta}_t M)(\alpha_t^2 C_0 + \beta_t^2 M)^{-1}(x - \beta_t m_1) \,,
$$

which matches the formula in the Gaussian setting before ($m_1 = 0$). $\qquad \diamond$

*Remark* C.3. Consider the 1D bimodal case

$$
\mu^*(x) = p \mathsf{N}(x; M, 1) + (1 - p) \mathsf{N}(x; -M, 1) \,.
$$

For general $\alpha_t, \beta_t$, using the formula in Proposition C.1, we have

$$
\begin{aligned}
b_t(x) = \dot{\beta}_t &\frac{p M \mathsf{N}(x; \beta_t M, \beta_t^2 + \alpha_t^2) - (1 - p) M \mathsf{N}(x; -\beta_t M, \beta_t^2 + \alpha_t^2)}{p \mathsf{N}(x; \beta_t M, \beta_t^2 + \alpha_t^2) + (1 - p) \mathsf{N}(x; -\beta_t M, \beta_t^2 + \alpha_t^2)} \\
&+ (\beta_t \dot{\beta}_t + \alpha_t \dot{\alpha}_t)(\beta_t^2 + \alpha_t^2)^{-1} \frac{p(x - \beta_t M) \mathsf{N}(x; \beta_t M, \beta_t^2 + \alpha_t^2) + (1 - p)(x + \beta_t M) \mathsf{N}(x; \beta_t M, \beta_t^2 + \alpha_t^2)}{p \mathsf{N}(x; \beta_t M, \beta_t^2 + \alpha_t^2) + (1 - p) \mathsf{N}(x; -\beta_t M, \beta_t^2 + \alpha_t^2)} \,.
\end{aligned}
\tag{C.10}
$$

Taking $\alpha_t = \sqrt{1 - \beta_t^2}$ leads to a simplified formula

$$
\begin{aligned}
b_t(x) &= \dot{\beta}_t \frac{p M \mathsf{N}(x; \beta_t M, \beta_t^2 + \alpha_t^2) - (1 - p) M \mathsf{N}(x; -\beta_t M, \beta_t^2 + \alpha_t^2)}{p \mathsf{N}(x; \beta_t M, \beta_t^2 + \alpha_t^2) + (1 - p) \mathsf{N}(x; -\beta_t M, \beta_t^2 + \alpha_t^2)} \\
&= \dot{\beta}_t M \frac{p \exp(2 \beta_t M x) - (1 - p)}{p \exp(2 \beta_t M x) + (1 - p)} = \dot{\beta}_t M \tanh(h + \beta_t M x)
\end{aligned}
$$

where $h$ satisfies $\frac{p}{1-p} = \exp(2h)$ or $p = \frac{\exp(h)}{\exp(h) + \exp(-h)}$.

Moreover, for the $d$ dimensional bimodal Gaussian mixture

$$
\mu^*(x) = p \mathsf{N}(x; r, \mathrm{I}) + (1 - p) \mathsf{N}(x; -r, \mathrm{I}) \,,
$$

a similar calculation implies $b_t(x) = \dot{\beta}_t r \tanh(h + \beta_t \langle r, x \rangle)$. $\qquad \diamond$

## C.3 Optimizing avg-Lip$^2$ for 1D Gaussian mixtures

*Proof for Proposition 3.6.* Using the formula in (3.7), we have $\nabla b_t(x) = M^2 \dot\beta_t \beta_t \operatorname{sech}^2(h + \beta_t M x)$ and

$$A_2 = \int_0^1 \mathbb{E}[\|\nabla b_t(I_t)\|_2^2]\,\mathrm{d}t = M^4 \int_0^1 \mathbb{E}[\dot\beta_t^2 \beta_t^2 \operatorname{sech}^4(h + \beta_t\, M I_t)]\mathrm{d}t\,. \tag{C.11}$$

We denote $G(u) = \mathbb{E}[\operatorname{sech}^4(h + uM(\sqrt{1-u^2}z + ux_1))]$, so $A_2 = M^4 \int_0^1 \dot\beta_t^2 \beta_t^2 G(\beta_t)\mathrm{d}t$. The Euler-Lagrange equation satisfies

$$\frac{\mathrm{d}}{\mathrm{d}t}\frac{\partial}{\partial\dot\beta_t}(\dot\beta_t^2\beta_t^2 G(\beta_t)) = \frac{\partial}{\partial\beta_t}(\dot\beta_t^2\beta_t^2 G(\beta_t))\,.$$

Using the Beltrami Identity, the equation leads to

$$\dot\beta_t^2\beta_t^2 G(\beta_t) - \dot\beta_t\frac{\partial}{\partial\dot\beta_t}(\dot\beta_t^2\beta_t^2 G(\beta_t)) = \text{const}\,,$$

which implies $\dot\beta_t^2\beta_t^2 G(\beta_t) = \text{const}$ and thus $\dot\beta_t\beta_t(G(\beta_t))^{1/2} = \text{const}$. Integrating both sides leads to the solution

$$t = \frac{\int_0^{\beta_t} u(G(u))^{1/2}\mathrm{d}u}{\int_0^1 u(G(u))^{1/2}\mathrm{d}u}\,.$$

Now, we derive the ODE that $\beta_t$ satisfies. To do so, we need to write out the integral over space explicitly. The density of $I_t$ satisfies

$$\rho_t(x) = p\mathsf{N}(x; \beta_t M, 1) + (1-p)\mathsf{N}(x; -\beta_t M, 1) = \frac{1}{\sqrt{2\pi}}\exp(-\frac{x^2 + \beta_t^2 M^2}{2})\frac{\cosh(h + \beta_t M x)}{\cosh(h)}\,.$$

Let us denote $\rho_t(x) = \rho(\beta_t, x)$ in this proof, which allows us to write

$$A_2 = M^4 \int_0^1 \int_{\mathbb{R}} L(\dot\beta_t, \beta_t, x)\rho(\beta_t, x)\mathrm{d}x\mathrm{d}t\,, \tag{C.12}$$

where $L(\dot\beta_t, \beta_t, x) = \dot\beta_t^2 \beta_t^2 \operatorname{sech}^4(h + \beta_t\, M x)$. The Euler-Lagrange equation for this problem has the form

$$\int_{\mathbb{R}}\left(\frac{\mathrm{d}}{\mathrm{d}t}\frac{\partial}{\partial\dot\beta_t}(L(\dot\beta_t, \beta_t, x)\rho(\beta_t, x))\right)\mathrm{d}x = \int_{\mathbb{R}}\left(\frac{\partial}{\partial\beta_t}(L(\dot\beta_t, \beta_t, x)\rho(\beta_t, x))\right)\mathrm{d}x\,.$$

We organize the equation according to $\rho$, which leads to

$$\int_{\mathbb{R}}(\frac{\partial}{\partial\beta_t}L - \frac{\mathrm{d}}{\mathrm{d}t}\frac{\partial}{\partial\dot\beta_t}L)\rho\mathrm{d}x = \int_{\mathbb{R}}(\frac{\partial}{\partial\dot\beta_t}L\frac{\mathrm{d}}{\mathrm{d}t}\rho - L\frac{\partial}{\partial\beta_t}\rho)\mathrm{d}x\,, \tag{C.13}$$

where we omit the arguments for simplicity of notation.

We have

$$\frac{\partial}{\partial\beta_t}L = 2\dot\beta_t^2\beta_t\operatorname{sech}^4(h + \beta_t M x) - 4Mx\dot\beta_t^2\beta_t^2\operatorname{sech}^4(h + \beta_t M x)\tanh(h + \beta_t M x)$$

$$\frac{\partial}{\partial\dot\beta_t}L = 2\dot\beta_t\beta_t^2\operatorname{sech}^4(h + \beta_t M x)$$

$$\frac{\mathrm{d}}{\mathrm{d}t}\frac{\partial}{\partial\dot\beta_t}L = (2\ddot\beta_t\beta_t^2 + 4\dot\beta_t^2\beta_t)\operatorname{sech}^4(h + \beta_t M x) - 8Mx\dot\beta_t^2\beta_t^2\operatorname{sech}^4(h + \beta_t M x)\tanh(h + \beta_t M x)$$

$$\frac{\mathrm{d}}{\mathrm{d}t}\rho = \dot\beta_t\frac{\partial}{\partial\beta_t}\rho = \dot\beta_t(-\beta_t M^2 + Mx\tanh(h + \beta_t M x))\rho$$

which shows that the left and right hand sides of (C.13) are

$$\text{LHS} = \int_{\mathbb{R}}\operatorname{sech}^4(h + \beta_t M x))\left(-2\dot\beta_t^2\beta_t - 2\ddot\beta_t\beta_t^2 + 4Mx\dot\beta_t^2\beta_t^2\tanh(h + \beta_t M x)\right)\rho\mathrm{d}x\,,$$

$$\text{RHS} = \int_{\mathbb{R}} ((\frac{\partial}{\partial \dot{\beta}_t} L) \dot{\beta}_t - L) \frac{\partial}{\partial \beta_t} \rho \mathrm{d}x = \int_{\mathbb{R}} \left( (2 \dot{\beta}_t \beta_t^2 \operatorname{sech}^4(h + \beta_t Mx)) \dot{\beta}_t - L \right) \frac{\partial}{\partial \beta_t} \rho \mathrm{d}x$$

$$= \int_{\mathbb{R}} \dot{\beta}_t^2 \beta_t^2 \operatorname{sech}^4(h + \beta_t Mx)) \left( -\beta_t M^2 + Mx \tanh(h + \beta_t Mx) \right) \rho \mathrm{d}x \,.$$

Since LHS = RHS, we get

$$\mathbb{E}[\left( -2 \dot{\beta}_t^2 \beta_t - 2 \ddot{\beta}_t \beta_t^2 + \dot{\beta}_t^2 \beta_t^3 M^2 + 3 \dot{\beta}_t^2 \beta_t^2 M I_t \tanh(h + \beta_t M I_t) \right) \operatorname{sech}^4(h + \beta_t M I_t)] = 0 \,.$$

Now, we note the fact that $\mathbb{E}[x \tanh(h + \beta_t M I_t)] = \beta_t M$ since

$$\mathbb{E}[I_t \tanh(h + \beta_t M I_t)] = \frac{1}{\sqrt{2\pi}} \int_{\mathbb{R}} e^{-\frac{x^2 + \beta_t^2 M^2}{2}} \frac{\cosh(h + \beta_t Mx)}{\cosh(h)} \tanh(h + \beta_t Mx) x \mathrm{d}x$$

$$= \frac{1}{\sqrt{2\pi} \cosh(h)} \int_{\mathbb{R}} e^{-\frac{x^2 + \beta_t^2 M^2}{2}} \sinh(h + \beta_t Mx) x \mathrm{d}x$$

$$= \frac{1}{2\sqrt{2\pi} \cosh(h)} \int_{\mathbb{R}} (e^h e^{-\frac{(x - \beta_t M)^2}{2}} - e^{-h} e^{-\frac{(x + \beta_t M)^2}{2}}) x \mathrm{d}x$$

$$= \frac{1}{\sqrt{2\pi}} \cdot \int_{\mathbb{R}} (\frac{e^h}{e^h + e^{-h}} e^{-\frac{(x - \beta_t M)^2}{2}} - \frac{e^{-h}}{e^h + e^{-h}} e^{-\frac{(x + \beta_t M)^2}{2}}) x \mathrm{d}x$$

$$= \frac{e^h}{e^h + e^{-h}} \beta_t M + \frac{e^{-h}}{e^h + e^{-h}} \beta_t M = \beta_t M \,.$$

Thus, we have

$$\mathbb{E}[I_t \tanh(h + \beta_t M I_t) \operatorname{sech}^4(h + m I_t)]$$

$$= \operatorname{Cov}(I_t \tanh(h + \beta_t M I_t), \operatorname{sech}^4(h + \beta_t M I_t)) + \mathbb{E}[I_t \tanh(h + \beta_t M I_t)] \mathbb{E}[\operatorname{sech}^4(h + \beta_t M I_t)]$$

$$= \operatorname{Cov}(I_t \tanh(h + \beta_t M I_t), \operatorname{sech}^4(h + \beta_t M I_t)) + \beta_t M \mathbb{E}[\operatorname{sech}^4(h + \beta_t M I_t)] \,.$$

With these formulas, the Euler-Lagrange equation becomes

$$-2 \dot{\beta}_t^2 \beta_t - 2 \ddot{\beta}_t \beta_t^2 + \dot{\beta}_t^2 \beta_t^3 M^2 (4 + 3 \operatorname{Corr}(I_t \tanh(h + \beta_t M I_t), \operatorname{sech}^4(h + \beta_t M I_t))) = 0 \,.$$

If we omit the Corr term, we get the ODE

$$\dot{\beta}_t^2 \beta_t - \ddot{\beta}_t \beta_t^2 + 2 \dot{\beta}_t^2 \beta_t^2 M^2 = 0 \,.$$

By setting $f_t = \beta_t^2$, the above ODE becomes $\ddot{f}_t = M^2 \dot{f}_t$. Solving this ODE with the correct boundary condition leads to

$$\beta_t = \frac{1}{M} \sqrt{-\log(-M^2 t + \frac{M^2}{1 - e^{-M^2}}) + \log \frac{M^2}{1 - e^{-M^2}}} \,,$$

which can be simplified as $\beta_t = \frac{1}{M} \sqrt{-\log(1 + (e^{-M^2} - 1)t)}$.

On the other hand, we note that if we optimize $\int_0^1 \mathbb{E}[\|\nabla b_t(I_t)\|_2^{2k}] \mathrm{d}t$, we will get

$$-\dot{\beta}_t^2 \beta_t - \ddot{\beta}_t \beta_t^2 + \dot{\beta}_t^2 \beta_t^3 2 M^2 \left( 1 + \frac{8k^2 - 6k + 1}{8k^2 - 4k} \operatorname{Corr}(I_t \tanh(h + \beta_t M I_t), \operatorname{sech}^{4k}(h + \beta_t M I_t)) \right) = 0 \,.$$

Omitting the correlation part leads to the same equation. Also, using the argument at the beginning of this proof, we have in such case

$$t = \frac{\int_0^{\beta_t} u(G(u))^{1/2k} \mathrm{d}u}{\int_0^1 u(G(u))^{1/2k} \mathrm{d}u} \,,$$

where $G(u) = \mathbb{E}[\operatorname{sech}^{4k}(h + uM(\sqrt{1 - u^2} z + u x_1))]$.

$\square$

# D   Experimental Details for Navier–Stokes

We give full details of the data, network, training, sampling and evaluation pipelines underlying the results in Section 4.4.

**Data generation and processing.** We integrate the stochastically forced Navier–Stokes vorticity equation (4.1) on the torus $\mathbb{T}^2 = [0, 2\pi]^2$ using a pseudo-spectral solver with explicit time stepping and standard 2/3 de-aliasing. Parameters are $\nu = 10^{-3}$, $\alpha = 0.1$, and $\varepsilon = 1$ as in the main text. Trajectories are run on a $256 \times 256$ spatial grid; after a long burn-in to reach the invariant regime, vorticity snapshots are saved at fixed time intervals. Five independent trajectories produce roughly $10^5$ snapshots in total. Each snapshot is normalized by a fixed empirical per-pixel norm computed once over the union of trajectories, so the resulting fields have unit standard deviation. At evaluation resolution $64 \times 64$, snapshots are downsampled by bilinear interpolation; at $128 \times 128$, snapshots are likewise interpolated from the native $256 \times 256$ grid. We split the dataset into 90% train and 10% test.

**Stochastic interpolant setup.** The source $z$ is a spatial Gaussian random field, sampled independently per training step. The target $x_1$ is a vorticity snapshot drawn from the training set. We use the linear interpolant $I_t = (1 - t)z + t\,x_1$ for training and learn the conditional drift $\mathbb{E}[\dot{I}_t \mid I_t]$ by minimizing the squared loss summed over channels and pixels and averaged over the batch and time. Times $t$ are sampled from a uniform distribution on $[10^{-3}, 1 - 10^{-3}]$.

**Network architecture.** The drift field is parameterized by a UNet Ho et al. (2020). We use base channels 32 with channel multipliers $(1, 2, 2, 2)$, ResNet block groups of size 8, and four downsampling/upsampling stages. Time conditioning uses a learned sinusoidal position embedding of dimension 32. Attention is applied at the lower-resolution stages with 4 heads and head dimension 32. The model has approximately 2,060,000 trainable parameters. The same architecture is used at resolutions $64 \times 64$ and $128 \times 128$, with input/output channel count one (vorticity is a scalar field). No class conditioning is used.

**Training.** The model is trained with AdamW at base learning rate $10^{-4}$, batch size 100, for 50,000 gradient steps. The learning rate follows a cosine annealing schedule. Gradients are clipped at norm $10^4$. Training is performed on a single GPU with mixed precision disabled.

**Sampling.** At inference time the ODE $\dot{z} = b_t(z)$ is integrated by a fixed-step fourth-order Runge–Kutta method on a uniform grid in $[t_{\min}, t_{\max}] = [10^{-3}, 1 - 10^{-3}]$. For an integration with $N$ grid points, $N - 1$ RK4 steps of equal size are taken. For the designed schedule, the drift is computed by the transfer formula derived in Section 3, which expresses the designed-schedule drift in terms of the trained linear-schedule drift and so requires no retraining. For each (schedule, step count, seed) triple we generate 500 samples from a fixed batch of initial conditions, and we use the same initial conditions across schedules within a seed for paired comparison.

**Choice of $\lambda^\star$.** The schedule parameter $\lambda^\star$ in (3.11) is selected automatically as in (4.2): the ratio of data to noise enstrophy spectra at the highest resolved wavenumber $k_{\max}$, computed once from the training set. This produces $\lambda^\star \approx 3 \times 10^{-4}$ at $64 \times 64$ and $\lambda^\star \approx 10^{-5}$ at $128 \times 128$. The procedure requires no manual tuning.

**Spectra and band errors.** For a 2D field $\omega$ of side $N$, we compute $\hat{\omega}(\mathbf{m}) = \mathcal{F}\omega$ and the radially-averaged enstrophy spectrum
$$S(k) \;=\; \pi(k_+^2 - k_-^2) \cdot \mathrm{mean}\big\{|\hat{\omega}(\mathbf{m})|^2 \;:\; |\mathbf{m}| \in [k_-, \, k_+)\big\},$$
where $(k_-, k_+) = (k - 0.5, k + 0.5)$ and $k$ ranges over integer wavenumbers from 1 to $N/2$. Spectra are averaged over the ensemble of generated samples before being compared with the test-set spectrum. Per-band relative errors are computed as the unweighted average of $|S_{\mathrm{gen}}(k) - S_{\mathrm{truth}}(k)|/|S_{\mathrm{truth}}(k)|$ over wavenumbers $k$ in each band; band cuts are $k < 8$, $8 \leq k < 24$, $k \geq 24$.

**Non-Gaussian metrics.** For a snapshot $\omega$ on the spatial lattice, structure functions are computed as
$$S_p(r) = \tfrac{1}{2}\,\mathbb{E}[|\omega(\cdot + re_1) - \omega(\cdot)|^p] + \tfrac{1}{2}\,\mathbb{E}[|\omega(\cdot + re_2) - \omega(\cdot)|^p],$$

averaging over both axes and all valid translations. The flatness is $F(r) = S_4(r)/S_2(r)^2$, with Gaussian baseline 3. The gradient kurtosis equals $F(1)$ (both expectations taken over the pixel-difference distribution). The Kolmogorov–Smirnov distance is computed between $10^5$ randomly sampled pixel values from the truth and the same number from the generated ensemble using the standard two-sample formula. All non-Gaussian statistics in Table 3 are reported as mean and standard deviation across five independent random seeds.

