# OpenReview forum: "Lipschitz-Guided Design of Interpolation Schedules in Generative Models"
_TMLR — Under review for TMLR_

### Review · Reviewer_4QZM · 2026-06-22

**Summary Of Contributions:**

This work presents a theoretical study of interpolation schedules within the stochastic interpolant framework. The authors make two main contributions.

First, they establish a statistical equivalence among scalar interpolation schedules by evaluating a path-KL criterion for marginal-preserving SDEs whose volatility coefficients are optimally tuned for each schedule. This result extends the equivalence shown by Kingma et al. [1] for diffusion-model objectives via SNR transformations. A consequence is that, in principle, it suffices to train a model under one interpolation schedule and then transfer it to another schedule through an appropriate time-change formula. The paper develops this idea in the deterministic setting, operating solely with ODE dynamics.

Second, and more centrally, the authors propose a criterion for selecting interpolation schedules that improves inference. Specifically, they minimize the time-averaged squared Lipschitz constant of the drift, measured through the averaged squared norm of its gradient. This contrasts with related Schrödinger bridge formulations, where schedule selection is often motivated by minimizing the squared drift norm itself. The authors compare the resulting schedules against the standard linear schedule. For Gaussian and Gaussian-mixture targets, the optimal schedules can be derived in tractable form and are shown to improve sampling accuracy under a limited number of ODE discretization steps. In more realistic settings, including Allen–Cahn and Navier–Stokes equations, the schedules are obtained from a Gaussian approximation of the target distribution and yield similar improvements over the linear interpolant.

[1] Kingma et al., Variational Diffusion Models, NeurIPS 2021.

**Audience:**

Yes

**Audience Explanation:**

The novelty of this paper lies in its proposal to optimize interpolation schedules for stochastic interpolants by minimizing the squared Lipschitz constant of the drift, rather than the squared norm of the drift itself. This criterion is well motivated, can be applied without retraining, and represents a potentially interesting contribution to the study of accelerated diffusion and flow-based generative models.

**Claims And Evidence:**

No

**Claims Explanation:**

Although I am convinced by the second part of the paper, namely the design of Lipschitz-optimal schedules, I am less convinced by the first contribution, which aims to establish an equivalence between interpolation schedules.

My main concern is that the empirical study is conducted entirely in the ODE setting, which is well motivated since ODE dynamics are argued to be more robust to discretization error. However, the proposed equivalence result is established in an SDE setting, by introducing volatility coefficients chosen optimally to minimize a KL divergence between an approximate and a ground-truth SDE. I find this ODE–SDE bridge somewhat artificial. In practice, the optimal volatility coefficients derived in the theory are not used; rather, the volatility is typically chosen to be constant or fixed by design.

This also makes the comparison with Kingma et al. [1] only partially convincing. In that work, the KL criterion is directly tied to the training objective through the variational bound. By contrast, in the stochastic interpolant framework, the training objective described in Section 2.2 is not directly based on the path-KL criterion used in the equivalence proof. As a result, the connection between the theoretical criterion and the actual learning procedure feels indirect.

Overall, I do not find the KL-based argument sufficient to support a meaningful practical equivalence between schedules. In my view, the same practical conclusion could be obtained more directly by observing that, once the drift has been learned under one interpolation schedule, one can transfer it to another schedule through the deterministic time-change formula in the ODE setting (Proposition 3.1). This would avoid introducing an artificial ODE–SDE bridge and would align more closely with the experiments and practical use case considered in the paper.

My other concerns are about the numerical evaluation and detailed below.

[1] Kingma et al., Variational Diffusion Models, NeurIPS 2021.

**Requested Changes:**

In my view, the main weakness of the paper lies in its numerical evaluation. The following changes are important for me to support acceptance.

**Major changes**

First, the proposed Lipschitz criterion is naturally opposed to the optimal transport kinetic criterion, as underlined by the authors. Although these settings do not rely on exactly the same ingredients—optimal transport baselines may require additional training, for instance through rectified flow procedures—the paper should compare them numerically. In particular, it would be useful to understand in which regimes the proposed criterion is preferable to OT-inspired alternatives, and vice versa. This comparison could be performed in the Gaussian and Gaussian-mixture settings, together with appropriate ablations. Such an experiment would substantially strengthen the motivation for introducing the new Lipschitz-based criterion.

Second, the evaluation on target distributions appears quite limited. As far as I understand, tractable optimal schedules can only be derived for Gaussian and Gaussian-mixture targets, and are then used as proxies for more complex targets. This proxy-based approach should be evaluated in a more scalable setting, such as image generation, which is a standard benchmark for diffusion- and flow-based generative models. For example, experiments at a resolution such as 256×256, with comparisons against schedules beyond the linear interpolant, would make the claims much more convincing. In its current form, the evaluation is mostly restricted to relatively simple settings that are not representative of the benchmarks typically used in the diffusion-model literature, which limits the demonstrated applicability of the method.

As explained above, I don't think that the statistical equivalence between interpolation schedules is meaningful and compatible with ODE experiments; hence, I think it should be revised.

**Medium changes**

Although the authors discuss the O(1)-step-size setting in Example 3.5 to argue a priori for the advantage of the optimal schedule over the linear schedule, the number of ODE steps used in Experiment 4.2, apparently chosen in accordance with the remark and ranging from 2 to 4, does not match a realistic sampling budget, which is typically closer to 20–30 function evaluations. As a result, the comparison between the two schedules in this experiment feels somewhat artificial, since such an extremely small number of steps is rarely used in practice. The number of function evaluations considered here should be aligned with the budgets used in the other experiments, for instance Experiment 4.1.

In the same Gaussian-mixture setting, the mode-recovery metric does not need to be computed using one-dimensional PCA. Since the target is a mixture of two unimodal Gaussians, it should be sufficient, and more direct, to use a posterior estimator based on the density ratio between the two Gaussian components.

---

### Review · Reviewer_bh9c · 2026-06-30

**Summary Of Contributions:**

Author study scalar interpolation schedules of $(\alpha_t, \beta_t)$ in diffusion/flow models within the stochastic interpolants framework. They study both from a statistical and numerical perspectives. Stats side, they show that (Proposition 2.5) KL div. in the path space is independent of these schedulers, which generalizes similar result for diffusion models in the literature. Numerical side, they propose to minimize the averaged squared Lipschitzness of the drift criteria. They show the transfer formulas between different schedulers given that we know drift of one scheduler. Then they derive optimal schedules for gaussians and gaussian mixtures, and log-concave targets. They provide numerical experiments on gaussian fields, 1000-dim gaussian mixture, and invariant measures of stochastic Allen–Cahn and Navier–Stokes equations.

The derivations I checked are correct and clean, including path-space equivalence, transfer formula, gaussian examples. Applications to the stochastic PDE invariant measures is interesting and motivational.

The downside is that main baseline is always the naive linear schedule, even if paper cites several previous works on schedulers. There is no rigorous connection provided between proposed criteria to choose scheduler (avg-Lip^2) and actual objective. Experiments are based on ODEs, but statistical equivalence proposition was proven for optimally tuned SDEs.

**Audience:**

Yes

**Audience Explanation:**

Yes, designing optimal schedulers is an active topic in diffusion/flow models. Viewing it in numerical side and proposing criteria for finding schedulers along with closed-form solutions and stochastic-PDE applications are within clear interest.

**Broader Impact Concerns:**

The Broader Impact Statement is present in the paper.

**Claims And Evidence:**

No

**Claims Explanation:**

- Most of the comparison claims (exponential improvement, fixed budget but more accurate, ...) are done against the linear schedule. It will be more convincing to add experiments against other schedulers mentioned in related works (EDM, dilated, learned, and others). Experiments basically show that proposed criteria A_2 is better than linear, not other strong methods.

- In Section 2, the equivalence proposition is proven for the setting that is not used in the rest of the paper. This equivalency holds after optimal tuning with $\epsilon_t = \alpha_t^2 (\frac{\dot \beta_t^2}{\beta_t} -  \frac{\dot \alpha_t^2}{\alpha_t})$, which is a specific SDE. But, experiments use the ODE setting with $\epsilon_t = 0$ in which the scheduler invariant accuracy was not shown. Moreover, the optimal $\epsilon_t$ is singular for $t$ approaching $0$, and equivalent SDE is shown to be well posed only for the linear case in Appendix B (general case was only mentioned).


Adding/fixing the baselines and further clarifications on section 2 can fix these problems.

**Requested Changes:**

**Critical**

- Add competitive baselines to the paper. Include dilated scheduler in the Table 1 comparisons. Moreover, add baselines for PDE experiments as well (like against standard ones - EDM, cosine, ... - and OT or learned schedulers). Because, paper currently is given as better than linear, which is not very convincing given that the field has plenty of strong scheduler designed.

- Can you please clarify that section 2 doesn't imply the equivalency for ODEs, which are used in the experiments?

- Can you please generalize the well posedness claim of SDE for general schedulers where $\epsilon_t$ is ill defined?

**Would simple strengthen**

- Can you add error bars/seeds for table 1 as well (similar to tables 2 and 3)?

- Can you extend experiments to the standard image datasets as claims are for general diffusion/flow models, but experiments are only on the PDEs and gaussians?

---

> ### Author Response · Authors · 2026-07-18
> **Reply and thank you for the detailed review**
>
> Thank you for the detailed review. One point we can clarify immediately is that the
> result in Section 2 concerns path-space KL for the SDE after optimizing its diffusion
> coefficient; it does not imply that the ODE discretizations used in the experiments are
> equivalent. The ODE experiments instead use the deterministic transfer formula in
> Proposition 3.1. We will make this separation explicit in the revision. We are also
> working on broader schedule baselines, a careful statement of the endpoint assumptions,
> seed and uncertainty reporting, and a CelebA image-generation experiment. We will
> upload the revised manuscript and our full response next week.

---

### Review · Reviewer_yPLP · 2026-07-03

**Summary Of Contributions:**

This paper investigates the interpolation schedule in the stochastic interpolant framework both theoretically and numerically. The authors prove that all scalar interpolation schedules are statistically equivalent under path-space KL divergence with certain chosen diffusion noise. They also propose avg-Lip2 objective, which is shown to be effective on both analytic and empirical solutions.

**Audience:**

Yes

**Audience Explanation:**

It interests people in both generative modeling and AI-driven PDE modeling areas.

**Claims And Evidence:**

Yes

**Claims Explanation:**

The submission contains valid proofs, analytic and empirical experiments, and visualizations.

**Requested Changes:**

1. The paper proposes minimizing avg-Lip2 as the design principle, but does not establish a rigorous link between this metric and actual numerical integration error bounds.
2. The experimental results on Allen–Cahn and Navier–Stokes invariant measures do not use the actual avg-Lip2-minimizing schedule for the true target distribution. Instead, they rely on a heuristic Gaussian proxy. This weakens the practicality of the new algorithm. The approximation error should be discussed. Plus, the analytic schedule for Gaussian mixtures omits a correlation term for tractability, yet it is never investigated. [critical]
3. Add competitive baselines to all experimental sections. There are many variants of SI/diffusion with different schedules. The paper should compare fairly and support its claim of optimality. [critical]
4. Evaluation on image generation benchmarks.

---

> ### Author Response · Authors · 2026-07-18
> **Reply and thank for the careful review**
>
> Thank you for the careful review. We agree with the main concerns, especially that the
> current paper does not establish a general link between avg-Lip\(^2\) and integration
> error, and that the Gaussian proxy needs better validation. We are revising those
> claims, checking the approximation used in the mixture example, and adding stronger
> schedule baselines and a CelebA image-generation experiment. We wanted to reply briefly
> now to let you know that this work is underway. We will upload the revised manuscript
> and a point-by-point response next week.